# A phosphate-binding pocket in cyclin B3 is essential for XErp1/Emi2 degradation in meiosis I

Rebecca Schunk [1,2], Marc Halder [1,2], Michael Schäfer [1], Elijah Johannes[1], Andreas Heim [1], Andreas Boland [3] & Thomas U Mayer [1,2 ✉]

## Abstract

To ensure the correct euploid state of embryos, it is essential that vertebrate oocytes await fertilization arrested at metaphase of meiosis II. This MII arrest is mediated by XErp1/Emi2, which inhibits the ubiquitin ligase APC/C (anaphase-promoting complex/cyclosome). Cyclin B3 in complex with Cdk1 (cyclin-dependent kinase 1) is essential to prevent an untimely arrest of vertebrate oocytes in meiosis I by targeting XErp1/Emi2 for degradation. Yet, the molecular mechanism of XErp1/Emi2 degradation in MI is not well understood. Here, by combining TRIM-Away in oocytes with egg extract and in vitro studies, we demonstrate that a hitherto unknown phosphate-binding pocket in cyclin B3 is essential for efficient XErp1/Emi2 degradation in meiosis I. This pocket enables Cdk1/cyclin B3 to bind pre-phosphorylated XErp1/Emi2 facilitating further phosphorylation events, which ultimately target XErp1/ Emi2 for degradation in a Plk1- (Polo-like kinase 1) dependent manner. Key elements of this degradative mechanism are conserved in frog and mouse. Our studies identify a novel, evolutionarily conserved determinant of Cdk/cyclin substrate specificity essential to prevent an untimely oocyte arrest at meiosis I with catastrophic consequences upon fertilization.

**Keywords** APC/C; Cyclin B3; Emi2; Phosphate-binding Pocket; XErp1
**Subject Categories** Cell Cycle; Post-translational Modifications & Proteolysis; Signal Transduction

## Introduction

Mitotic and meiotic cell cycle transitions are mediated by the phosphorylation of distinct substrates by cyclin-dependent kinases (Cdks) in complex with activating cyclin subunits (Crncec and Hochegger, 2019; Hochegger et al, 2008; Kamenz and Ferrell, 2017). The rise and fall in the activity of a distinct Cdk/cyclin complex is primarily determined by the synthesis and destruction of the respective cyclin subunit. In higher eukaryotes, A- and B-type cyclins in complex

with Cdk1 orchestrate entry into mitosis and meiosis, while progression through these cell cycle phases is primarily driven by Cdk1/cyclin B (Fung and Poon, 2005; Hégarat et al, 2020; Hochegger et al, 2008; Vigneron et al, 2018). Within the family of B-type cyclins, cyclin B1 and cyclin B2 are highly conserved in their amino acid sequences. In contrast, cyclin B3, initially discovered in a chicken cDNA library (Gallant and Nigg, 1994), forms a distinct subfamily with higher sequence conservation among cyclin B3 proteins from divergent species than with cyclin B1 or B2 from the same species (Fig. EV1A) (Nguyen et al, 2002). Notably, mammalian cyclin B3 is roughly three times larger than its orthologue in non-mammalian vertebrates due to an enormous extension of exon 8 (Lozano et al, 2012). Mutations in human Ccnb3, the gene encoding cyclin B3, are associated with recurrent triploid pregnancies and miscarriage (Fatemi et al, 2021; Rezaei et al, 2022; Wang et al, 2023). Mice lacking cyclin B3 (Ccnb3$^{-/-}$) are viable with fertile males, but infertile females (Karasu et al, 2019; Karasu and Keeney, 2019; Li et al, 2019b; Zhang et al, 2015). Ccnb3$^{-/-}$ oocytes undergoing meiotic maturation fail to complete the first meiotic division, but arrest at metaphase of meiosis I (MI) with high Cdk1 activity and unseparated bivalents. Under physiological conditions, immature oocytes are arrested at prophase of meiosis I (prophase I arrest) until hormonal stimulation triggers them to undergo meiotic maturation, i.e., they complete the first meiotic division and then arrest as mature eggs at metaphase of meiosis II (MII), where they await fertilization (Jessus et al, 2020). This MII arrest, also termed CSF (cytostatic factor) arrest, is mediated by XErp1/Emi2, which inhibits the ubiquitin ligase APC/C (anaphase promoting complex/cyclosome) and thereby prevents degradation of the anaphase inhibitors cyclin B and securin (Liu and Maller, 2005; Madgwick et al, 2006; Schmidt et al, 2005; Shoji et al, 2006; Tung et al, 2005). Fertilization triggers degradation of XErp1/Emi2, thereby resulting in APC/C activation and, hence, in exit from MII (Hansen et al, 2006; Liu et al, 2006; Rauh et al, 2005). Recently, we dissected cyclin B3's essential function in female meiosis by demonstrating that it targets the APC/C inhibitor XErp1/Emi2 for proteasomal degradation in meiosis I (Bouftas et al, 2022). This mechanism, which is conserved from frog to mouse and includes Plk1 (Polo-like kinase 1), is essential to keep the levels of XErp1/Emi2 during meiosis I below the threshold critical for APC/C inhibition (Bouftas et al, 2022; Meng et al, 2020). Destruction of cyclin B3 during exit from MI allows XErp1/Emi2 accumulation resulting in APC/C inhibition essential for entry into MII.

[1]Department of Biology, University of Konstanz, 78457 Konstanz, Germany. [2]Konstanz Research School Chemical Biology, University of Konstanz, Universitätsstraße 10, 78457 Konstanz, Germany. [3]Department of Molecular and Cellular Biology, University of Geneva, 30 Quai Ernest-Ansermet, 1211 Geneva, Switzerland. ✉E-mail: thomas.u.mayer@uni-konstanz.de

Cyclins, in addition to their canonical function as activators of Cdks, can exert additional tasks (Hydbring et al, 2016). A prime example is vertebrate cyclin B1, which in complex with Cdk1 acts as a stoichiometric inhibitor of separase (Gorr et al, 2005; Stemmann et al, 2001; Yu et al, 2023). Separase, upon APC/C-mediated degradation of both its inhibitors securin and cyclin B, cleaves the cohesin ring encircling sister chromatids and thereby triggers chromosome segregation at anaphase onset (Konecna et al, 2023). A prerequisite for complex formation between Cdk1/cyclin B1 and separase, which results in mutual inhibition of both enzymes, is phosphorylation of separase at serine residue 1126 (pS1126) (Gorr et al, 2005; Stemmann et al, 2001). Recently, the structure of human Cdk1 associated with the phospho-threonine adaptor protein Cks1 and cyclin B1 (Cdk1/Cks1/cyclin B1, CCC) bound to separase was determined by cryogenic electron microscopy (cryoEM) (Yu et al, 2021). This study identified a hitherto unknown phosphate-binding pocket (PBP) in cyclin (cyc) B1 with arginine 307 (R307), histidine 320 (H320), and lysine 324 (K324) forming a hydrogen-bonding network with the phosphate group attached to separase S1126 (Fig. 1A). Critical residues of the positively charged phosphate-binding pocket are conserved within B-type cyclins across species including yeast, but no other cyclin families (Fig. 1B). Thus, this PBP defines a novel specificity site present exclusively in B-type cyclins. The absence of this pocket in A-type cyclins explains why cyclin B1 and B2, but not cyclin A2, bind to and inhibit separase (Li et al, 2019a; Touati et al, 2012). Likewise, separase with a non-phosphorylatable S1126A mutation is resistant to cyclin B1-mediated inhibition (Stemmann et al, 2001). Of note, the mechanism of mutual inhibition of separase and Cdk1/cyclin B is restricted to vertebrates (Hellmuth et al, 2015), suggesting that the pocket evolved originally to fulfill separase-independent functions. Indeed, for yeast it was recently reported that multisite phosphorylation of the transcriptional co-activator Ndd1 by Cdk1 seems to involve both Cks1 and Clb2's phosphate-binding pocket (Asfaha et al, 2022).

Notably, the positive charges of cyclin B1's phosphate-binding pocket (R302, K324) are conserved by two arginine residues in cyclin B3, while H320 is replaced by a leucine residue (Fig. 1B). All three residues are highly conserved in cyclin B3 across a broad range of species. We therefore speculated that cyclin B3 contains a functional phosphate-binding pocket that allows Cdk1/cyclin B3 to bind pre-phosphorylated substrates facilitating sequential multisite substrate phosphorylation. Such a pocket-dependent substrate anchoring mechanism could enhance phosphorylation of low-affinity sites leading to high occupancy substrate phosphorylation and contribute to ultrasensitive switches in protein function and substrate specificity. As shown previously (Trunnell et al, 2011), switch-like responses in protein function are critical for a timely and ordered execution of cell cycle processes (Kamenz and Ferrell, 2017).

## Results

### Cyclin B3 possesses a functional phosphate-binding pocket

To test if cyclin B3 possesses a functional phosphate-binding pocket, we first performed immunoprecipitation (IP) experiments to analyze if cyclin B3 interacts with separase, the first reported pocket-dependent interactor of cyclin B1. For these experiments, we used extract prepared from *Xenopus* eggs naturally arrested at metaphase of meiosis II, so-called CSF (cytostatic factor) extract. Since cyclin B3 is not expressed in MII (Bouftas et al, 2022), CSF extract allows the functional analysis of ectopic cyclin B3 without interfering with endogenous cyclin B3. CSF extract was supplemented with in vitro transcribed and translated (IVT) Flag-tagged wildtype (WT) *Xenopus* cyclin B1 or cyclin B3 followed by an α-Flag IP. Since ectopic cyclin B3 induces CSF extract to exit meiosis (Bouftas et al, 2022), extracts were supplemented with inhibitors of the proteasome (MG262) and of protein phosphatases 1 and 2 (okadaic acid, OA) to ensure that all samples were arrested at MII. As expected, endogenous separase efficiently co-precipitated with Flag-cyclin B1$^{WT}$ (Fig. 1C). In contrast, separase was not detectable in the cyclin B3 IP sample despite efficient immunoprecipitation of Flag-cyclin B3$^{WT}$. From these data, we speculated that either cyclin B3 does not possess a functional phosphate-binding pocket or that its pocket interacts with phosphorylated proteins other than separase. To test the latter idea, we performed an α-Flag IP from CSF extract supplemented with IVT *Xenopus* Flag-cyclin B3$^{WT}$, MG262 and OA and analyzed the IP samples using a pan-specific phospho-threonine antibody (pThr Ab). As control, we performed an α-Flag IP from CSF extract not supplemented with Flag-cyclin B3. Indeed, compared to the control IP, the pThr signal was significantly stronger in the cyclin B3$^{WT}$ IP beads sample (Fig. 1D). α-Flag-cyclin B3$^{WT}$ IP followed by treatment with lambda phosphatase (WT + λ) strongly reduced the pThr signal of bead samples, confirming that the antibody shows specificity towards phosphorylated proteins. To confirm that cyclin B3's phosphate-binding pocket mediates binding to these phosphorylated proteins, we created a pocket-mutant (PM) version of cyclin B3. Specifically, we mutated R296, L309, and R313 to the corresponding residues in human cyclin A2 (R296T, L309E, R313M, Fig. 1B). WB analyses of the α-Flag IP from CSF extract supplemented with IVT Flag-cyclin B3$^{PM}$ revealed that the pThr signal in the IP samples was significantly reduced especially for proteins with higher molecular weight (Fig. 1D). We also performed WB with a pan-specific phospho-serine antibody, which showed a similar pattern as the pThr Ab (Fig. EV1B). From these experiments, we concluded that cyclin B3 contains indeed a functional phosphate-binding pocket that interacts with phosphorylated proteins.

Next, we sought to confirm that the introduced mutations did not unspecifically affect cyclin B3 function. To this end, we performed radiometric ([γ-$^{33}$P]ATP) in vitro kinase assays using recombinant His-tagged Cdk1 in complex with either WT or PM Strep-cyclin B3 (Strep-cyclin B3$^{WT/PM}$) and histone H1 as a generic substrate. Autoradiography analyses revealed that histone H1 was efficiently phosphorylated by Cdk1/cyclin B3$^{WT}$ in the DMSO solvent control reaction (Fig. 1E,F). When the Cdk inhibitor flavopiridol was added, no signal was detected confirming that histone H1 phosphorylation was mediated by Cdk1. Importantly, in the presence of the solvent control DMSO, Cdk1/cyclin B3$^{PM}$ was as efficient as Cdk1/cyclin B3$^{WT}$ in phosphorylating histone H1 (Fig. 1E,F) demonstrating that the introduced mutations did not unspecifically affect cyclin B3 function. In sum, these data demonstrated that cyclin B3 has a functional phosphate-binding pocket that does not interact with separase, but with other pre-phosphorylated proteins.

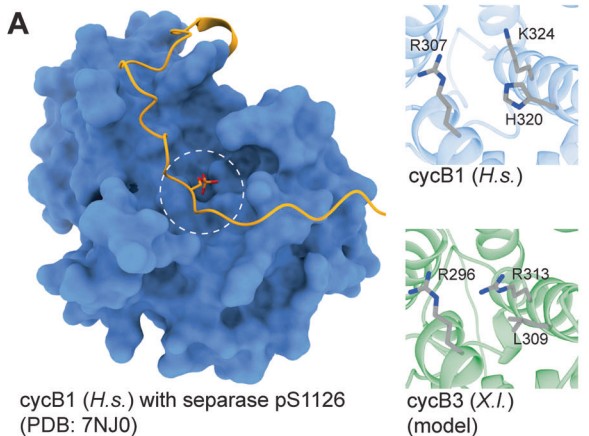

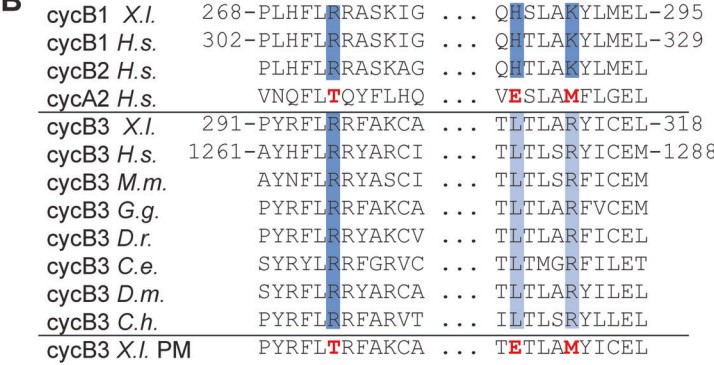

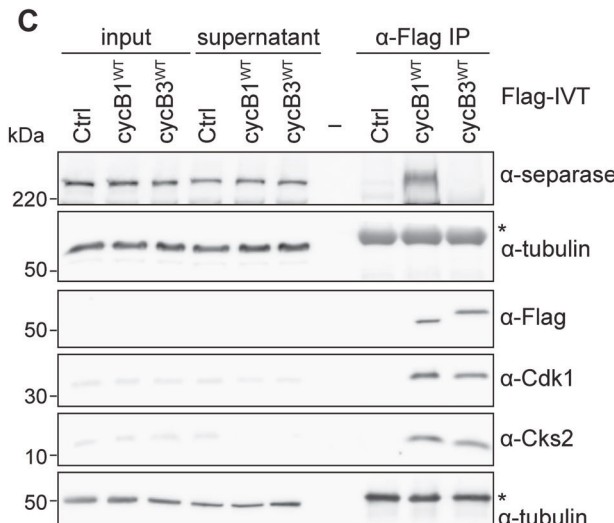

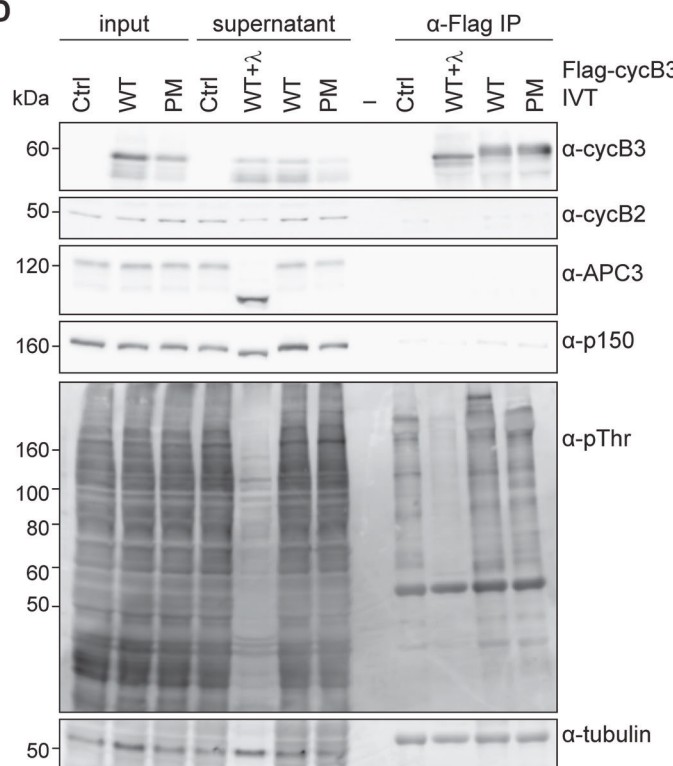

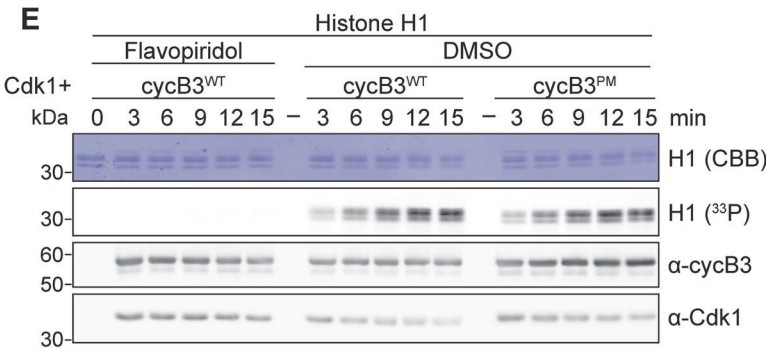

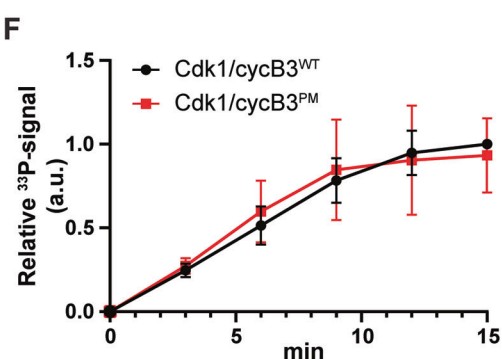

◄ **Figure 1. Cyclin B3 contains a functional phosphate-binding pocket.**

(A) Structure of human (*H.s.*) cyclin B1 in complex with separase phosphorylated on S1126 (PBD: 7NJ0). White circle depicts region of the phosphate-binding pocket. Images on the right show the phosphate-binding pocket of human cyclin B1 with R307, H320, and K324 (top) and the corresponding residues in *Xenopus* (*X.l.*) cyclin B3 in close-up. (B) Sequence alignments of B-type cyclins (cyc) from different species and human cyclin A2 in the region of the phosphate-binding pocket in cyclin B1. Marked in dark blue are the three critical residues in human cyclin B1 (R307, H320, K324) forming the hydrogen-bonding network with separase pS1126 and the corresponding conserved residues in other B-type cyclins. Light blue marks the residues in cyclin B3 corresponding to H320 and K324. Non-conserved residues in human cyclin A2 are marked in red. Last row shows the sequence of pocket mutant (PM) cyclin B3 with the three critical residues mutated to the corresponding residues of human cyclin A2 (R296T, L309E, R313M). *X.l.* (*Xenopus laevis*), *H.s.* (*Homo sapiens*), *M.m.* (*Mus musculus*), *G.g.* (*Gallus gallus*), *D.r.* (*Danio rerio*), *C.e.* (*Caenorhabditis elegans*), *D.m.* (*Drosophila melanogaster*), *C.h.* (*Clytia hemisphaerica*). Residue numbers are labelled on the right. (C) Western blot (WB) analysis of α-Flag immunoprecipitation (IP) samples from CSF extract supplemented with IVT WT Flag-cyclin B3 or Flag-cyclin B1. To ensure that the extract remained arrested at metaphase II, MG262 and okadaic acid were added. α-tubulin served as a loading control. One representative experiment from three independent biological replicates is shown. Asterisk marks heavy chain of α-Flag Ab used for the IP. (D) WB analysis of α-Flag IP from CSF extract supplemented with IVT Flag-cyclin B3$^{WT/PM}$. The meiotic state of the extract was maintained by the addition of MG262 and okadaic acid. Indicated samples were treated with λ-phosphatase ($+ λ$). p150 served as a loading control. One representative experiment from three independent biological replicates is shown. (E) Radiometric ([γ-$^{33}$P]ATP) in vitro kinase assay using recombinant His-Cdk1 in complex with either WT or PM Strep-cyclin B3, and histone 1 (H1) as substrate. As indicated, reactions were performed in the presence of the solvent control DMSO or Flavopiridol. Samples were taken at indicated time points and analyzed by Coomassie brilliant blue staining (CBB), WB and autoradiography ($^{33}$P). One representative experiment from four independent replicates is shown. (F) Quantification of experiment shown in (E) and three further technical replicates. The $^{33}$P signal was quantified and normalized to Cdk1 levels detected by WB (3 min time point). Mean ± SD is depicted ($n = 4$). Source data are available online for this figure.

## Oocytes expressing pocket mutant cyclin B3 prematurely arrest at metaphase I

Next, we sought to understand the physiological function of cyclin B3's phosphate-binding pocket. Vertebrate oocytes lacking cyclin B3 prematurely arrest at metaphase of meiosis I due to an untimely accumulation of the APC/C inhibitor XErp1 (Bouftas et al, 2022). To test if cyclin B3's phosphate-binding pocket contributes to the essential function of cyclin B3 in meiosis I, we depleted endogenous cyclin B3 from prophase I-arrested oocytes using TRIM-Away and expressed in these oocytes either WT or mutant cyclin B3 followed by progesterone (PG) treatment to induce meiotic maturation (Fig. 2A). TRIM-Away utilizes the ubiquitin ligase TRIM21, which binds antibodies (Ab) and targets them together with bound antigen for proteasomal degradation (Clift et al, 2017). To deplete cyclin B3 from immature oocytes, we co-injected them with mRNA encoding TRIM21 and a cyclin B3 peptide antibody (cycB3$^{Ab}$), which was previously validated (Bouftas et al, 2022). WB analyses confirmed efficient depletion of endogenous cyclin B3 from oocytes co-injected with TRIM21 mRNA and cycB3$^{Ab}$, compared to oocytes co-injected with TRIM21 mRNA and an unspecific IgG control Ab (Fig. 2B). For the detection of cyclin B3 we used in this experiment, as in all others, an antibody that was raised against an N-terminal fragment of cyclin B3 (aa 1–110). Under physiological conditions, immature oocytes resume meiosis upon PG stimulation by activating Cdk1/cyclin B and after completing the first meiotic division arrest as fertilizable eggs at metaphase of meiosis II. In *Xenopus* oocytes, the appearance of a white spot in the oocyte's animal hemisphere caused by pigment dispersal at the cell cortex as a consequence of germinal vesicle breakdown (GVBD) serves as morphological readout for PG-induced Cdk1 activation. According to our experience, four hours after GVBD most oocytes have completed MI and progress towards MII or are already arrested at metaphase II. Based on the exemplary images shown in Fig. 2C, we categorized the meiotic stage of the oocytes under the different experimental conditions as "MI" and "post-MI". As expected, immunofluorescence analyses of oocytes four hours post GVBD revealed that the majority of cyclin B3-depleted oocytes failed to exit MI as they displayed MI spindles and no extruded polar bodies (PBs) (Fig. 2D). In most control-depleted oocytes, anaphase I

spindles or extruded polar bodies, with or without nearby metaphase II spindles, were observed. This suggested that these oocytes were either in the process of completing meiosis I (MI) or have already successfully completed it. To confirm that this effect was due to cyclin B3 depletion we performed rescue experiments by co-injecting IVT WT Flag-cyclin B3 (Δaa 1–15, cyclin B3$^{Δ15\ WT}$) that lacks part of the antigen region and, therefore, was not recognized by cycB3$^{Ab}$ (Bouftas et al, 2022). Indeed, most oocytes co-injected with TRIM21 mRNA, cycB3$^{Ab}$ and Flag-cyclin B3$^{Δ15\ WT}$ were able to complete MI (Fig. 2D). In stark contrast almost all oocytes co-injected with pocket mutant Flag-cyclin B3$^{Δ15\ PM}$ failed to complete MI (Fig. 2D). Reportedly, oocytes lacking cyclin B3 arrest in meiosis I due to precocious accumulation of XErp1 resulting in untimely APC/C inhibition (Bouftas et al, 2022). Thus, to confirm our IF analyses, we performed WB analysis of cyclin B3- and control-depleted oocytes before and after PG stimulation. Upon PG treatment, control-depleted oocytes resumed meiosis evident by the phosphorylation-dependent upshift of APC3 at GVBD. Four hours later (GVBD + 4 h), these oocytes were in the process of progressing towards MII, as indicated by the intermediate SDS-PAGE mobility of APC3, which was between the slow-migrating form observed one hour post-GVBD and the fast-migrating form of prophase I-arrested oocytes (Fig. 2E, compare lanes 4 with 3 and 5). Consistently, cyclin B2 was partially degraded in these oocytes at GVBD + 4 h (lane 4). As expected, in control-depleted oocytes XErp1 was not detectable until GVBD + 4 h, i.e., the time point when oocytes progressed towards MII (Fig. 2E, lane 4). Of note, samples were treated with lambda (λ)-phosphatase prior to SDS-PAGE as indicated to simplify the precise determination of XErp1 protein levels. Cyclin B3-depleted oocytes were able to resume meiosis but then arrested in MI evident by constantly upshifted APC3 and stable cyclin B2 (lanes 6–8). Notably, in cyclin B3-depleted oocytes XErp1 was already slightly detectable at GVBD (0 h) and strongly accumulated thereafter (Fig. 2E, lanes 6–8). Premature accumulation of XErp1 was prevented by the expression of IVT Flag-cyclin B3$^{Δ15\ WT}$, but not of cyclin B3$^{Δ15\ PM}$ (Fig. 2E, lane 11 vs 15). Consequentially, cyclin B3-depleted oocytes expressing the WT rescue construct completed the first meiotic division as shown by the intermediate SDS-PAGE mobility of APC3 and degradation of cyclin B2 at GVBD + 4 h (lane 12). From these data,

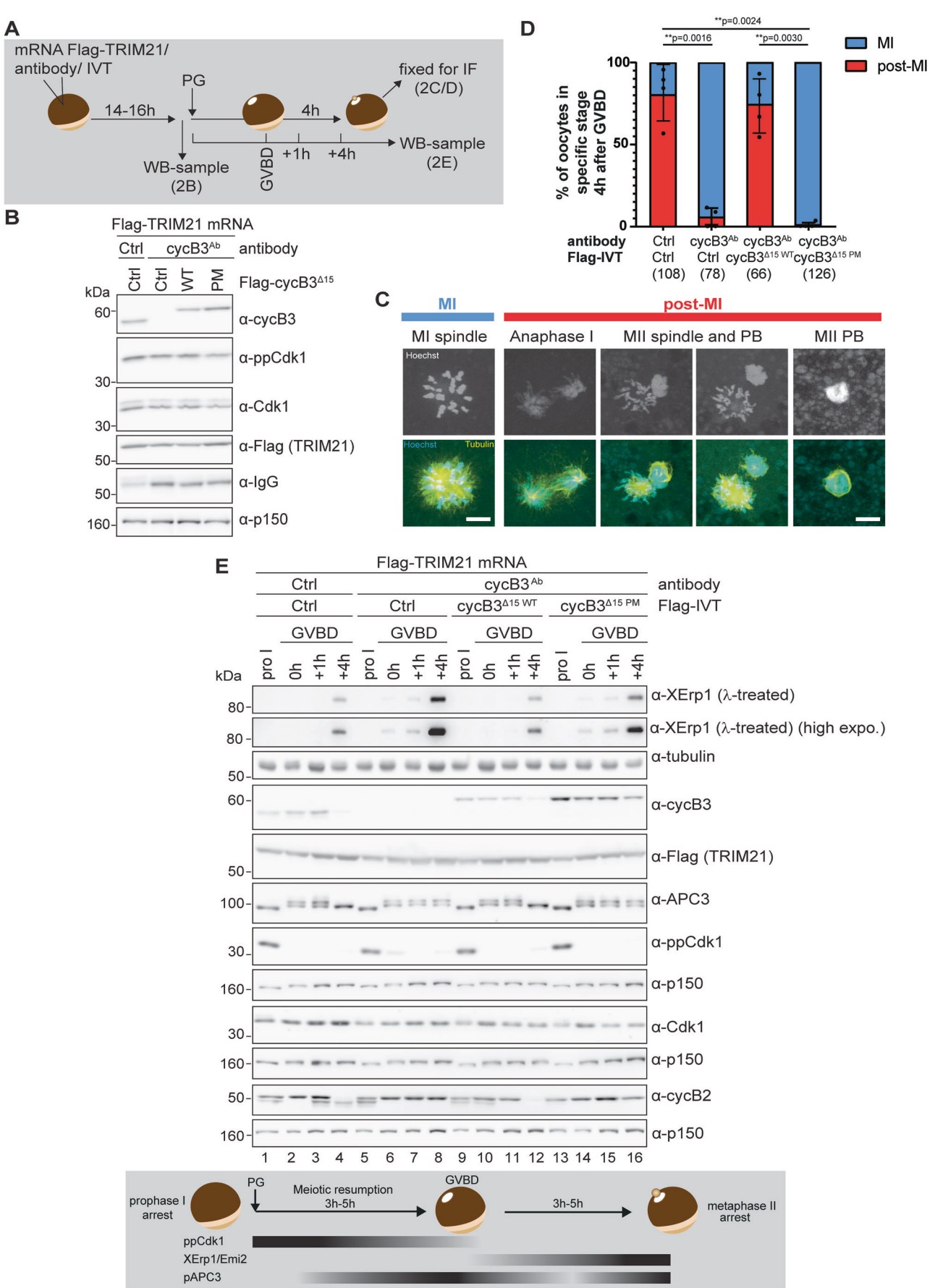

**Figure 2. Cyclin B3's essential function in meiosis I requires a functional phosphate-binding pocket.**

(A) Upper panel: Experimental outline of cyclin B3 depletion and rescue experiments in *Xenopus* oocytes. Immature oocytes were co-injected with TRIM21 mRNA and a peptide antibody targeting the N-terminus of cyclin B3 (aa 6–22, CycB3$^{Ab}$) or unspecific control (Ctrl) antibody. Where indicated, IVT of WT or pocket mutant cyclin B3$^{\Delta15}$ was co-injected. Cyclin B3$^{\Delta15}$ lacks the first 15 residues and therefore is not recognized by CycB3$^{Ab}$. Empty IVT (Ctrl) served as control. Oocytes were treated with progesterone (PG) or not and at indicated time points processed for immunofluorescence (IF) microscopy or WB analyses. (B) Oocytes injected as indicated were collected before progesterone (PG) treatment and processed for immunoblotting. Cyclin B3 was detected with an antibody raised against an N-terminal fragment of cyclin B3 (aa 1–110). p150 served as loading control. (C) Representative sample images of oocytes treated with PG and processed for immunofluorescence (IF) confocal microscopy four hours post GVBD. GVBD was determined by the appearance of a white spot in the oocyte's animal hemisphere. DNA and microtubules were visualized with Hoechst and FITC-labelled α-tubulin antibody, respectively. According to DNA and spindle morphologies, oocytes were classified in MI and post-MI groups and these categories were used for the quantification shown in (D). Scale bar: 10 μm. (D) Quantification of oocytes treated as described in (A). Number of experiments: $n = 4$, the total number of oocytes imaged is indicated under each condition (N). An unpaired two-sided *t* test with Welch's correction was performed, single data points and mean values are depicted. Error bars show SD, **$P < 0.01$. (E) WB analyses of oocytes treated as follows: Oocytes co-injected with TRIM21 mRNA, indicated antibodies and rescue constructs or control IVT (Ctrl) were treated with PG. At GVBD, 1 h or 4 h post GVBD oocytes were harvested and immunoblotted. Samples were treated with lambda (λ)-phosphatase as indicated. One representative experiment from six biological replicates is shown. Lower panel: Scheme of progesterone (PG)-induced meiotic progression of *Xenopus* oocytes. Disappearance of Cdk1 inhibitory phosphorylation (ppCdk1), accumulation of XErp1 and activating phosphorylation of APC3 are used to determine the meiotic cell cycle stage of oocytes. Source data are available online for this figure.

we concluded that cyclin B3's positively charged pocket is essential for its function in targeting the APC/C inhibitor XErp1 for degradation during meiosis I.

## Pocket integrity is important for XErp1 degradation, but not T97 phosphorylation

To investigate the molecular mechanism by which cyclin B3's phosphate-binding pocket contributes to XErp1 degradation, we analyzed the stability of endogenous XErp1 in CSF extract supplemented with IVT Flag-cyclin B3. As before, samples were treated with lambda (λ)-phosphatase as indicated to allow precise determination of XErp1 protein levels. Addition of IVT cyclin B3$^{WT}$ to CSF extract caused rapid degradation of XErp1 resulting in APC/C activation evident by cyclin B2 degradation (Fig. 3A). Subsequently, the extract exited MII, indicated by an increased SDS-PAGE mobility of APC3 due to its dephosphorylation. Cyclin B3's ability to destabilize XErp1 was dependent on its interaction with Cdk1 because expression of a hydrophobic patch mutant (cyclin B3$^{MRL}$)—shown to be deficient in Cdk1 binding (Bouftas et al, 2022)—had no effect on XErp1 stability (Fig. 3A). Importantly, in the presence of cyclin B3$^{PM}$, XErp1 was stable and consequentially the extract remained arrested at metaphase II, evident by stable cyclin B2 and sustained phosphorylation of APC3. Reportedly, mouse oocytes lacking Cks2 fail to complete the first meiotic division and arrest at metaphase I (Spruck et al, 2003), a phenotype reminiscent of Ccnb3$^{-/-}$ oocytes. To analyze if Cks2 is involved in XErp1 degradation, we depleted Cks2 from CSF extract supplemented with IVT cyclin B3$^{WT}$. Like in control depleted extract, IVT cyclin B3$^{WT}$ efficiently induced XErp1 degradation in extract depleted of Cks2 (Fig. EV1C,D). From these experiments, we concluded that cyclin B3's phosphate-binding pocket is an important determinant for the destruction of XErp1.

Next, we sought to better understand the mechanism of Cdk1/cyclin B3-mediated degradation of XErp1. Phosphorylation of XErp1 at threonine 97 (T97) by Cdk1/cyclin B3 is a prerequisite for efficient degradation of XErp1 (Fig. 3B) (Bouftas et al, 2022). Specifically, Cdk1/cyclin B3 directly phosphorylates T97, an evolutionary highly conserved site (Fig. 3C), and this phosphorylation event contributes to the recruitment of Polo-like kinase 1 (Plk1) to XErp1. Recruited Plk1 then phosphorylates XErp1 at two phospho-degrons (DSGX$_3$S$^{38}$, DSAX$_2$S$^{288}$). Upon phosphorylation

of the phospho-degrons by Plk1, XErp1 is recognized by the ubiquitin ligase SCF$^{\beta TRCP}$ resulting in efficient proteasomal degradation of XErp1 (Fig. 3B). Given the importance of T97 phosphorylation for XErp1 degradation, we first analyzed if wild-type and pocket mutant Cdk1/cyclin B3 differ in their ability to phosphorylate T97. To this end, we performed radiometric ([γ-$^{33}$P] ATP) in vitro kinase assays using recombinant *Xenopus* His-Cdk1 in complex with Strep-cyclin B3$^{WT/PM}$ and as substrate an N-terminal fragment of XErp1 (aa 1–350, XErp1$^{NT}$), tagged with maltose-binding protein (MBP). In CSF extract, XErp1$^{NT}$ was targeted for degradation by the addition of ectopic cyclin B3$^{WT}$ as efficiently as full-length (FL) XErp1 indicating that it contained all residues critical for its degradation (Fig. EV2A). To specifically analyze T97 phosphorylation, we mutated all of the eight Cdk1 consensus (S/T)P sites in XErp1$^{NT}$ (Isoda et al, 2011)—except for T97—to non-phosphorylatable sites (XErp1$^{NT\ 7A}$, Fig. EV2C).

Notably, autoradiography analyses revealed that Cdk1/cyclin B3$^{PM}$ was as efficient as the WT complex in phosphorylating XErp1$^{NT\ 7A}$ (Figs. 3D and EV2B). To confirm that T97 is the main phosphorylation site, we repeated the assay with XErp1$^{NT\ 8A}$ carrying an additional non-phosphorylatable mutation at T97. Indeed, this fragment was much less phosphorylated (Fig. 3D). From these data, we concluded that pocket integrity—while being important for XErp1 degradation (Fig. 3A)—does not contribute to T97 phosphorylation.

## Phosphorylated T97 engages cyclin B3's phosphate-binding pocket

To understand how cyclin B3's pocket contributes to XErp1 degradation, we first used AlphaFold2 (Evans, 2022; Jumper et al, 2021) to model *Xenopus* FL cyclin B3 in complex with XErp1. For these computational predictions, we used the same N-terminal fragment of XErp1 (XErp1$^{NT}$) that served as substrate for the in vitro kinase assay. Notably, the highest-ranking models positioned XErp1 T97 right next to R296 and R313, and in close proximity to L309 (Fig. 4A). The structure prediction then allowed the modelling of phosphorylated T97 near the pocket (using distance restraints to avoid steric clashes). Importantly, this model provides a structural rationale for the recognition of the phosphate group by the two conserved arginines in cyclin B3, which are well positioned to directly interact with the phosphate group of pT97.

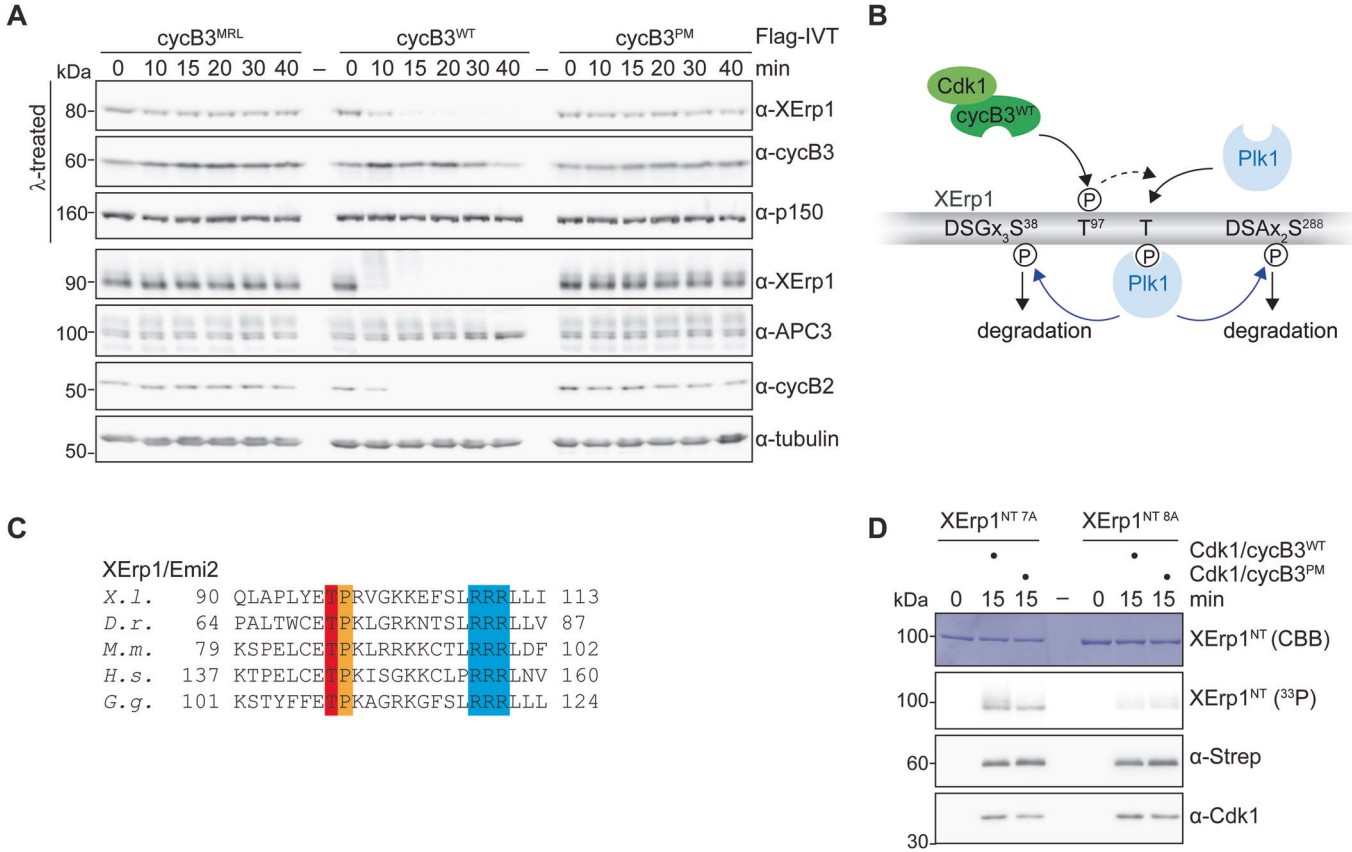

**Figure 3. Integrity of cyclin B3's pocket is important for the degradation of XErp1, but not for the phosphorylation of T97.**

(A) At indicated time points after supplementing CSF extract with IVT of Flag-tagged wildtype (WT), pocket mutant (PM), or hydrophobic patch mutant (MRL) cyclin B3, samples were taken, treated with lambda phosphatase (λ) where indicated and immunoblotted. p150 and α-tubulin served as loading control for the blots shown above. One representative experiment from three biological replicates is shown. (B) Scheme of Cdk1/cyclin B3-mediated degradation of XErp1. See text for details. (C) Sequence alignment of XErp1 from different species. *Xenopus* T97, P98 and triple arginine stretch (aa 108–110) as well as corresponding residues in XErp1/Emi2 from different species are labeled in red, orange and blue, respectively. *X.l.* (*Xenopus laevis*), *H.s.* (*Homo sapiens*), *M.m.* (*Mus musculus*), *G.g.* (*Gallus gallus*), *D.r.* (*Danio rerio*). (D) Radiometric ([γ-³³P]ATP) in vitro kinase assay using recombinant His-Cdk1/Strep-cyclin B3$^{WT/PM}$ and as substrate MBP-XErp1$^{NT}$ with all eight Cdk1 consensus sites mutated to alanine (8A) or seven non-phosphorylatable mutations and wildtype for T97 (7A). Samples were taken at indicated time points and analyzed by Coomassie brilliant blue staining (CBB), WB and autoradiography (³³P). One representative experiment from three independent replicates is shown. Source data are available online for this figure.

To test the hypothesis that phosphorylated T97 is directly recognized by cyclin B3's phosphate-binding pocket, we performed pull-down assays using phosphopeptides. Specifically, short peptides containing phosphorylated T97 (pT97) or not (T97) (CQLAPLYE(p)TPRVGKKE) were coupled to beads and then incubated in CSF extract supplemented with mRNA of WT or pocket mutant Flag-cyclin B3. Furthermore, to maintain the CSF arrest, extracts were supplemented with MG262 and IVT of a C-terminal fragment of XErp1 (XErp1$^{CT}$, aa 491–651), which, as shown previously (Bouftas et al, 2022), inhibits the APC/C but lacks residues mediating XErp1 degradation. Empty beads served as negative control. WB analyses of bead-bound fractions revealed that cyclin B3$^{WT}$ efficiently bound to the pT97 peptide, but not the T97 peptide or empty bead control (Fig. 4B). In contrast, cyclin B3$^{PM}$ was hardly detectable in the pT97 bead fraction and absent when incubated with the T97 peptide or control beads. To test if cyclin B3's PBP has a preference for phosphorylated threonine over serine residues, we repeated the pull-down assay using the same

peptide except that we replaced pT97 with pS97. Intriguingly, cyclin B3$^{WT}$ was undetectable in the pS97 bead fraction (Fig. EV3A) suggesting that cyclin B3—at least in the context of the tested peptide—preferentially binds to phosphorylated threonine residues. In sum, these data demonstrate that pocket integrity is essential for the interaction with phosphorylated T97.

To confirm the finding that phosphorylated T97 binds to the phosphate-binding pocket of cyclin B3 in the context of FL XErp1, we performed α-Flag IP experiments using CSF extract supplemented with IVT Flag-cyclin B3$^{WT/PM}$ and FL Myc-XErp1. α-Flag IP from extract lacking Flag-cyclin B3 served as control. IP experiments were performed using stable XErp1 (XErp1$^s$) variants (Fig. EV2C) and extracts were supplemented with MG262 and IVT XErp1$^{CT}$ to maintain their meiotic state. Analyses of the α-Flag IP samples revealed that Myc-XErp1$^{s\ WT}$ co-precipitated with cyclin B3$^{WT}$ but was absent in the control IP bead samples (Fig. 4C, lanes 7 and 9). No interaction between XErp1$^s$ and cyclin B3 was detected when T97 or the pocket was mutated (lanes 10 and 11), further supporting the idea

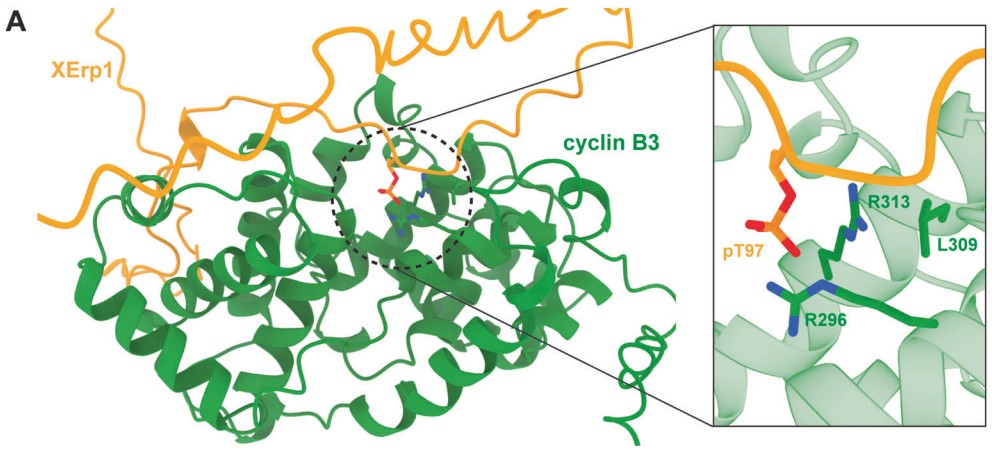

**A**

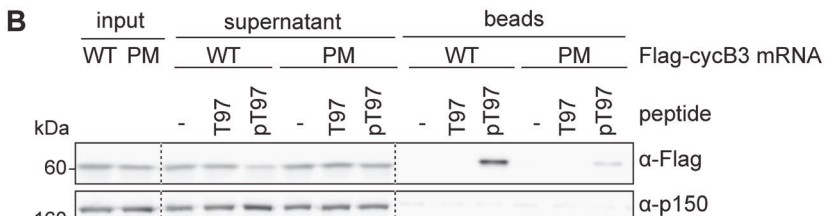

**B**

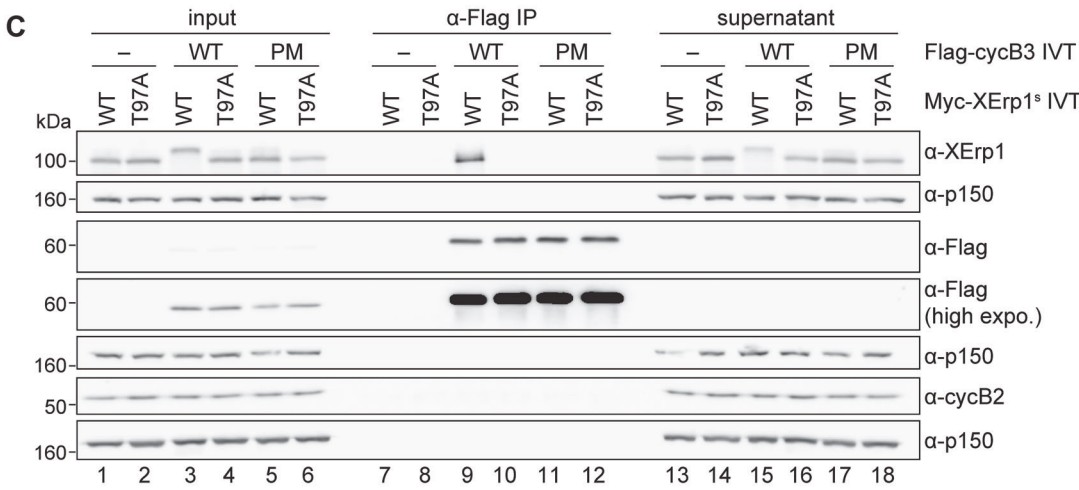

**C**

**D**

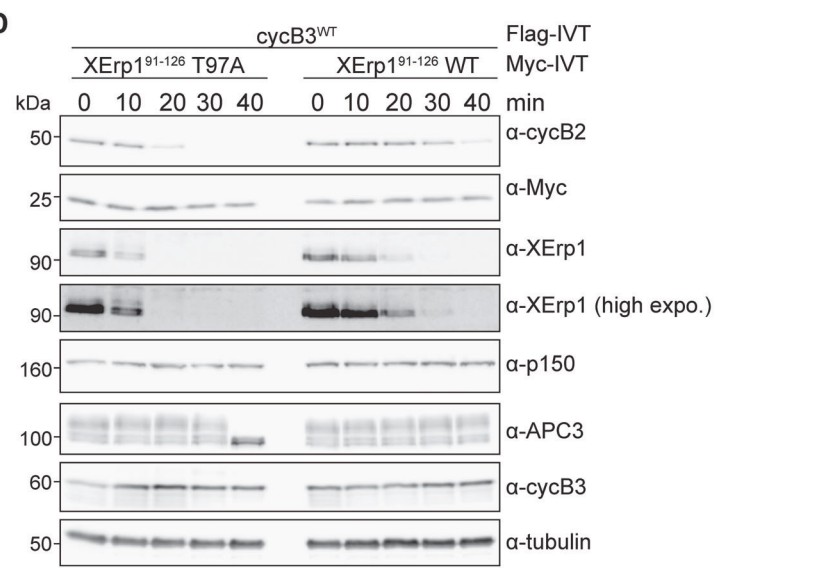

**Figure 4. Cyclin B3's phosphate-binding pocket engages phosphorylated T97.**

(A) AlphaFold-Multimer prediction of *Xenopus* cyclin B3 with XErp1$^{NT}$ (aa 1–350) was used to model phosphorylated T97 near the pocket. For clearer visualization, cyclin B3 residues 1–132 were removed. Cyclin B3 and XErp1$^{NT}$ are shown in green and orange, respectively. The black circle marks the region of the phosphate-binding pocket which is depicted in a close-up view in the right panel. (B) Bead-coupled peptides (XErp1 aa 90–104) comprising phosphorylated T97 (pT97) or not (T97) were incubated in CSF extract supplemented with mRNA encoding Flag-cyclin B3$^{WT/PM}$. Exit from MII was prevented by the addition of MG262 and IVT XErp1$^{CT}$. Input, supernatant, and beads were immunoblotted. p150 served as loading control. One representative experiment from three biological replicates is shown. (C) CSF extract was supplemented with IVT Flag-cyclin B3$^{WT/PM}$ and Myc-tagged XErp1$^{s\ WT/T97A}$. XErp1$^{s}$ is stabilized due to mutation of the two phospho-degrons (see EV2C). Exit from MII was prevented by MG262 and IVT XErp1$^{CT}$. Following α-Flag IP, input, supernatant and IP samples were immunoblotted. IP samples were treated with λ-phosphatase. p150 served as a loading control for the blots shown above. One representative experiment from four biological replicates is shown. (D) CSF extract was supplemented with IVT Myc-XErp1 fragment (aa 91–126), which was either WT or T97A. Of note, T97 is the only Cdk1 site within the fragment. Next, IVT Flag-cycB3$^{WT}$ was added and samples were taken at indicated time points. Samples were immunoblotted as indicated. α-tubulin and p150 served as loading control for the blots shown above, respectively. One representative experiment from three biological replicates is shown. Source data are available online for this figure.

that pT97 binds to cyclin B3's phosphate-binding pocket. Note that XErp1$^{s\ WT}$, but not XErp1$^{s\ T97A}$ underwent a phosphorylation-dependent shift in its SDS-PAGE mobility in the presence of cyclin B3$^{WT}$ but not of cyclin B3$^{PM}$ (Fig. 4C, lanes 3–5). Thus, since WT and mutant cyclin B3 are equally efficient in T97 phosphorylation (Fig. 3D), but not in pT97 binding (Fig. 4B,C) these data suggested that the interaction between pT97 and the WT pocket is important for further phosphorylation events. If docking of cyclin B3's phosphate-binding pocket to pT97 is critical for XErp1 degradation, we speculated that an ectopically expressed fragment of XErp1 comprising phosphorylated T97 should act as decoy titrating away cyclin B3 from endogenous XErp1. In consequence, degradation of endogenous XErp1 should be slowed down under these conditions. To test this idea, CSF extract was supplemented with IVT of a Myc-tagged XErp1 fragment comprising residues 91–126 and cyclin B3$^{WT}$. Of note, T97 is the only Cdk1 site in XErp1$^{91–126}$. As control, we used the same XErp1 fragment carrying a non-phosphorylatable T97A mutation. Indeed, degradation of endogenous XErp1 was significantly slowed down in the presence of the WT XErp1 fragment, compared to the condition when the extract was supplemented with the non-phosphorylatable XErp1 fragment (Fig. 4D). Consequentially, meiotic exit was impaired in the presence of WT XErp1$^{91–126}$ evident by continuous slow SDS-PAGE mobility of APC3 and delayed degradation of cyclin B2.

From these experiments, we concluded that a docking event mediated by pT97 and cyclin B3's phosphate-binding pocket is critical for XErp1 degradation.

## XErp1 is sequentially phosphorylated in a pT97-dependent manner

Based on the obtained results, we postulated that XErp1 degradation relies on a pocket-dependent phosphorylation event downstream of T97 phosphorylation. To test the idea that T97 phosphorylation is important for further phosphorylation events, we in vitro phosphorylated MBP-XErp1$^{NT\ WT}$, which was WT for all phosphorylation sites (Fig. EV2C), and used altered SDS-PAGE mobility as a proxy for the phosphorylation status. For these radiometric ([γ-$^{33}$P]ATP) in vitro kinase assays, we used recombinant Cdk1/cyclin B3$^{WT/PM}$ and MBP-XErp1$^{NT}$. XErp1$^{NT\ WT}$ displayed strongly retarded SDS-PAGE mobility after phosphorylation by Cdk1/cyclin B3$^{WT}$, but not when incubated with Cdk1/cyclin B3$^{PM}$ (Fig. 5A). Similarly, XErp1$^{NT\ T97A}$, which was WT for all phosphorylation sites except for T97, was also not upshifted when incubated with Cdk1/cyclin B3$^{WT}$. Of note, despite the fact that XErp1$^{NT}$ was not upshifted under these two conditions, it was still

radiolabeled indicating that it was also phosphorylated in a T97- and PBP-independent manner. In sum, these data showed that XErp1$^{NT\ T97A}$ carrying a single non-phosphorylatable site at T97 displayed the same SDS-PAGE mobility after phosphorylation by Cdk1/cyclin B3$^{WT}$ as XErp1$^{NT\ WT}$ phosphorylated by Cdk1/cyclin B3$^{PM}$. These data support our hypothesis that XErp1 undergoes sequential phosphorylation events that depend on both the integrity of the phosphate-binding pocket and T97. A corollary of this hypothesis is that phosphorylation of T97 is necessary but not sufficient for XErp1 degradation. To test this, we analyzed in CSF extract supplemented with IVT cyclin B3$^{WT}$ the stability of IVT full-length XErp1 (Fig. EV2C). Of note, we used XErp1*, which, as shown previously (Schmidt et al, 2005), was deficient in APC/C inhibition due to mutation of its zinc-binding region (Fig. EV2C), to exclude that ectopic XErp1 interferes with the cyclin B3-mediated exit from MII. As expected, FL XErp1*$^{WT}$, wildtype for all eight N-terminal Cdk1 sites (Fig. EV2C), was efficiently degraded in the presence of cyclin B3$^{WT}$ and mutation of these eight consensus sites (8A) strongly stabilized FL XErp1* (Fig. 5B). Importantly, mutation of just T97 (T97A) was sufficient to stabilize XErp1* demonstrating that T97 phosphorylation is required to destabilize XErp1. As aforementioned, Plk1 recruitment to XErp1 depends on T97 phosphorylation by Cdk1/cyclin B3. To analyze the role of T97 phosphorylation and pocket integrity in Plk1 recruitment, we performed co-IP experiments from CSF extract supplemented with IVT cyclin B3 and FL XErp1 variants (XErp1$^{s}$) with non-functional phosphodegrons to prevent their degradation by ectopic cyclin B3 (Fig. EV2C). Furthermore, we added MG262 and IVT XErp1$^{CT}$ to the extract to maintain the CSF arrest. In the presence of cyclin B3$^{WT}$, endogenous Plk1 co-precipitated with WT XErp1$^{s}$, but not with T97A XErp1$^{s}$ (Fig. 5C). Similarly, no interaction with Plk1 was observed when the extract was supplemented with WT XErp1$^{s}$ and pocket mutant cyclin B3. Since pocket integrity is not required for T97 phosphorylation (Fig. 3D), these data suggest that additional, pocket-dependent phosphorylations are necessary for efficient Plk1 recruitment. Consistently, T97, while being required for Plk1 recruitment, was not sufficient to destabilize Myc-XErp1*$^{7A}$, which was WT for T97 but carried non-phosphorylatable mutations at the remaining seven Cdk1 consensus sites (Figs. 5B and EV2C). Thus, despite the fact that Myc-XErp1*$^{7A}$ is WT for T97, T170 (pT170 serves as docking site for Plk1, see below), and the two phosphodegrons, it is not targeted for degradation by Cdk1/cyclin B3. From these experiments, we concluded that phosphorylation of XErp1 T97 by Cdk1/cyclin B3 is required to target XErp1 for degradation because it is a

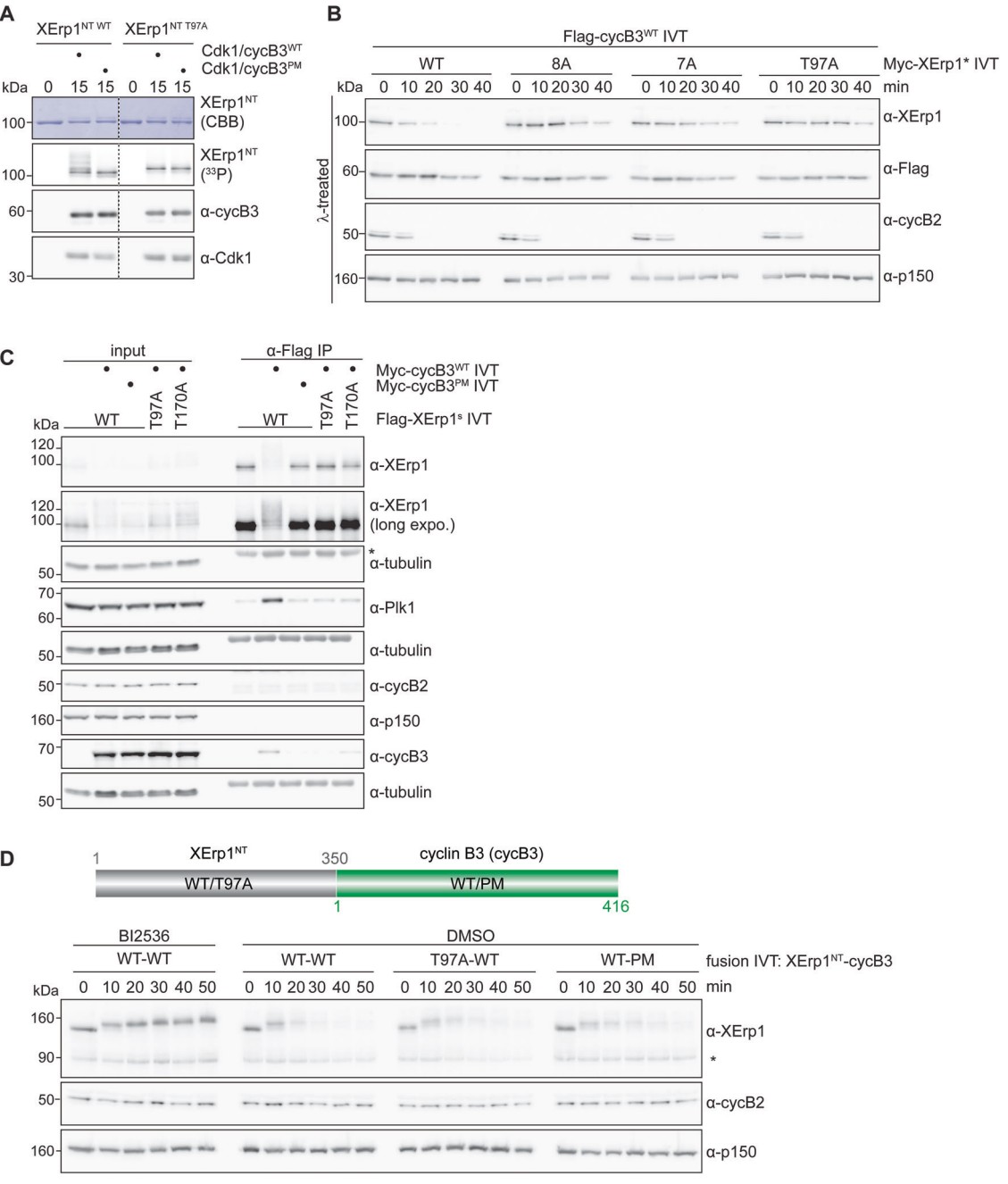

**Figure 5. T97 is critical for subsequent phosphorylation events resulting in XErp1 degradation.**

(A) Radiometric ([γ-³³P]ATP) in vitro kinase assay using recombinant His-Cdk1/Strep-cyclin B3$^{WT/PM}$ and as substrate WT or T97A MBP-XErp1$^{NT}$. Samples were taken at indicated time points and analyzed by Coomassie brilliant blue staining (CBB), WB and autoradiography (³³P). One representative experiment from three independent replicates is shown. (B) CSF extract was supplemented with IVT of the indicated Myc-tagged XErp1* constructs and Flag-cyclin B3$^{WT}$. XErp1* is deficient in APC/C inhibition (see EV2C) to prevent ectopic XErp1 from interfering with meiotic exit. At indicated time points, samples were taken and immunoblotted. p150 served as loading control. One representative experiment from three independent replicates is shown. (C) WB analysis of α-Flag IP from CSF extract supplemented with IVT Flag-XErp1$^s$ WT, T97A or T170A and Myc-cyclin B3$^{WT/PM}$. The meiotic state of the extract was maintained by the addition of MG262 and IVT of an APC/C-inhibitory C-terminal fragment of XErp1 (XErp1$^{CT}$). p150 served as a loading control. One representative experiment from three independent biological replicates is shown. Asterisk marks heavy chain of antibody used for IP. (D) Upper panel: Scheme of the fusion constructs comprising WT or T97A Myc-XErp1$^{NT}$ (aa 1–350, XErp1$^{NT\ WT/T97A}$) fused to full-length (FL) WT or pocket mutant cyclin B3 (cycB3$^{WT/PM}$). Lower panel: CSF extract was supplemented with IVT of the indicated Myc-tagged fusion constructs and the solvent control DMSO or BI2536. Exit from MII was prevented by the addition of IVT XErp1$^{CT}$. At indicated time points, samples were taken and immunoblotted. p150 served as loading control. One representative experiment from three biological replicates is shown. Asterisk marks endogenous XErp1. Source data are available online for this figure.

prerequisite for the recruitment of Plk1, but it is not sufficient to destabilize XErp1 because phosphorylations at additional Cdk1 (S/T)P sites are necessary. Previous studies have shown that Plk1 phosphorylates T170 by itself, thereby creating its own binding sites on XErp1 (Bouftas et al, 2022; Isoda et al, 2011). Building on these published data, we performed co-IP experiments from CSF extract supplemented with IVT cyclin B3$^{WT}$ and FL XErp1 T170A. Indeed, endogenous Plk1 failed to co-precipitate with XErp1 T170A, like it was the case with XErp1 T97A (Fig. 5C). Consistently, IVT XErp1 T170A was stable in CSF extract supplemented with IVT cyclin B3$^{WT}$ (Fig. EV3B). Thus, the degradation of XErp1 depends on two docking events with Cdk1/cyclin B3 docking to phosphorylated T97, and Plk1 to phosphorylated T170.

## Fusion of XErp1 to cyclin B3 makes pT97 and pocket integrity dispensable

Based on our data, we speculated that the docking event between pT97 and cyclin B3's pocket should become dispensable if cyclin B3 and XErp1 are artificially tethered to each other. To test this, we analyzed the stability of IVT Myc-XErp1$^{NT}$ fused at its C-terminus to FL cyclin B3 (XErp1$^{NT}$-CycB3) in CSF extract. To prevent that the fusion triggers APC/C activation and, thus, meiotic exit by targeting endogenous XErp1 for degradation, we added IVT XErp1$^{CT}$ to the extract. WB analyses revealed that XErp1$^{NT}$-cycB3$^{WT}$ quickly underwent phosphorylation-dependent SDS-PAGE mobility shifts followed by its efficient degradation (Fig. 5D). Of note, XErp1$^{NT}$-cycB3$^{WT}$ was stable in CSF extract treated with the Plk1 inhibitor BI2536 confirming that degradation of the fusion protein was mediated by the canonical pathway depending on Plk1 activity. After having validated our degradation assay, we analyzed the stability of fusion proteins carrying mutations at T97 or the pocket. Indeed, XErp1$^{NT\ T97A}$-cycB3$^{WT}$ was efficiently degraded confirming that T97 phosphorylation becomes dispensable when XErp1$^{NT}$ and cyclin B3 are fused together (Fig. 5D). Consequentially, XErp1$^{NT}$-cycB3$^{PM}$ was also efficiently degraded in CSF extract. In sum, these experiments support our hypothesis that docking of XErp1 pT97 to cyclin B3's pocket is critical for further destabilizing phosphorylations.

## Cyclin B3 and cyclin B1 differ in their pocket specificity

XErp1 levels are high during the CSF arrest of mature oocytes at metaphase of meiosis II, despite highly active Cdk1/cyclin B1 (Hansen et al, 2007; Wu and Kornbluth, 2008). However, we previously demonstrated that Cdk1/cyclin B1 is less efficient in T97 phosphorylation than Cdk1/cyclin B3 (Bouftas et al, 2022) providing an explanation of how XErp1 can accumulate to high levels in MII despite the presence of Cdk1/cyclin B1. Considering the critical role of cyclin B3's phosphate-binding pocket in destabilizing XErp1, we wondered if the pockets of cyclin B1 and cyclin B3 differ in their ability to bind XErp1 pT97. To test this, we repeated the phosphopeptide pull-down assay in CSF extract supplemented with mRNA encoding Flag-tagged WT cyclin B1 or cyclin B3. As shown before (Fig. 4B), cyclin B3$^{WT}$ efficiently bound to the pT97 peptide, which was also reflected by its significant depletion from the pT97 supernatant fraction (Fig. 6A). In strong contrast, cyclin B1$^{WT}$ was hardly detectable in the pT97 bead samples, and this was not due to reduced expression of cyclin B1

(Fig. 6A, input). Consistently, cyclin B3$^{WT}$ was much more efficient in binding pT97 than cyclin B1$^{WT}$ when both cyclins were present. Thus, cyclin B1 and cyclin B3 differ in their specificity for pocket-dependent substrates with cyclin B1 and cyclin B3 binding to separase and XErp1, respectively. To better understand the cause for this differential substrate specificity, we re-analyzed the AlphaFold model for cyclin B3 in complex with XErp1$^{NT}$. Notably, the area surrounding the phosphate-binding pocket in cyclin B3 is enriched for acidic residues (Fig. 6B), which are modeled to be in close proximity with a cluster of conserved basic residues (R108, R109, R110) in XErp1 (Figs. 3C and 6B). To test if these basic residues in XErp1 contribute to cyclin B3 binding, we performed co-IP experiments from CSF extract containing Myc-tagged XErp1 variants and Flag-cyclin B3$^{WT}$. Meiotic exit was prevented by the addition of MG262 and IVT XErp1$^{CT}$. As expected, WT but not T97A XErp1 was detected in the Flag-cyclin B3 IP (Fig. 6C). Importantly, XErp1 carrying three arginine to alanine mutations (3RA: R108A, R109A, R110A) also failed to co-precipitate with cyclin B3 suggesting that these basic residues contribute to cyclin B3 binding. Since cyclin B3 binding is a prerequisite for XErp1 degradation (Figs. 3–5), XErp1$^{3RA}$ should be stabilized compared to XErp1$^{WT}$ in the presence of cyclin B3$^{WT}$. Importantly, WB analysis confirmed that this was indeed the case (Fig. 6D). Notably, AlphaFold modeling of the same region in cyclin B1 revealed an oppositely charged surface in the proximity of the phosphate-binding pocket (Fig. EV3C). In sum, these data suggested that the differences in the residues surrounding the phosphate-binding pocket account for differential pocket specificity and, thus, for the fact that XErp1 is almost completely stable in the presence of cyclin B1, but rapidly degraded when cyclin B3 is present.

## The requirement of cyclin B3's pocket is evolutionarily conserved

Next, we analyzed if the mechanism of phosphate-binding pocket-dependent degradation of XErp1 is evolutionarily conserved. Reportedly, mouse oocytes—like Xenopus oocytes—lacking cyclin B3 arrest at metaphase of meiosis I due to untimely accumulation of Emi2, the mouse orthologue of XErp1 (Bouftas et al, 2022). Mouse Emi2 is phosphorylated at T86 (Xenopus T97) by Cdk1/cyclin B3, and this phosphorylation event contributes to the recruitment of Plk1, which then phosphorylates a single phosphodegron resulting in SCF$^{βTRCP}$-mediated degradation of Emi2 (Figs. 3B and 7A). Notably, AlphaFold2 modeling placed T86 of mouse Emi2 near the three conserved pocket residues R1273, L1286, and R1290 (Fig. 7B) of mouse cyclin B3. The structure prediction then allowed the modelling of phosphorylated T86 near the pocket (using distance restraints to avoid steric clashes) at an appropriate distance that allows the formation of a hydrogen-bonding network, as observed for the X. laevis complex (Fig. 4A). To create pocket mutant mouse cyclin B3, we mutated R1273, L1286, and R1290 to the corresponding residues (R1273→T, L1286→E, and R1290→M) in human cyclin A2 (Fig. 1B). To test if pocket integrity of mouse cyclin B3 is important for the degradation of mouse Emi2, Xenopus CSF extract was supplemented with mRNA of mouse Flag-cyclin B3 and IVT of Myc-Emi2, which was deficient in APC/C inhibition (Emi2*). Ectopic Emi2*$^{WT}$ was efficiently degraded in CSF extract supplemented

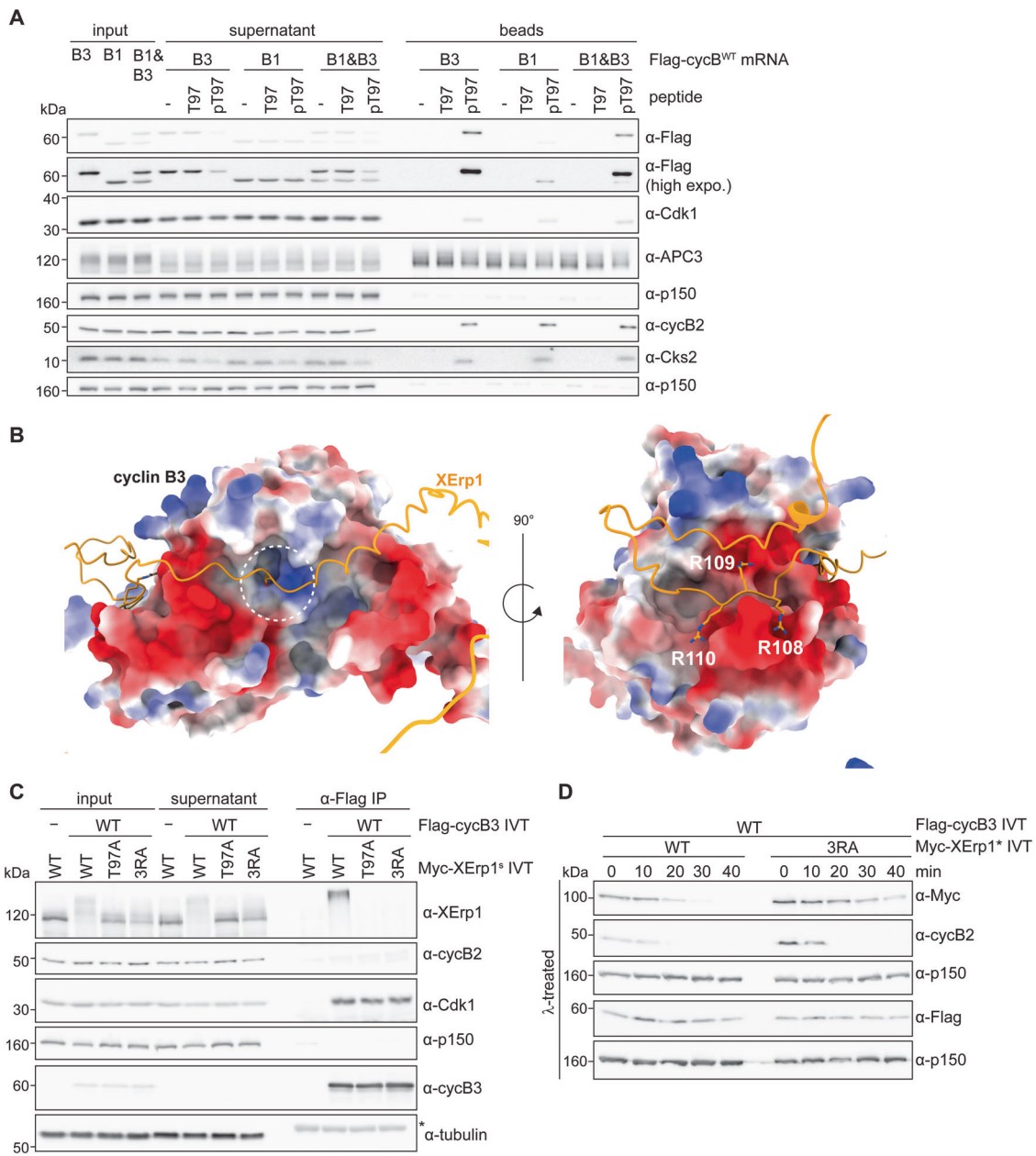

**Figure 6. An arginine cluster C-terminal to T97 is critical for XErp1 degradation.**

(A) Bead-coupled peptides (XErp1 aa 90–104) comprising phosphorylated T97 (pT97) or not (T97) were incubated in CSF extract supplemented with mRNA of WT Flag-cyclin B3 and/or Flag-cyclin B1. Exit from MII was prevented by the addition of MG262 and IVT XErp1$^{CT}$. Input, supernatant, and beads were immunoblotted. p150 served as loading control. One representative experiment from three biological replicates is shown. (B) Surface charge representation of AlphaFold-Multimer prediction of *Xenopus* cyclin B3 with XErp1$^{NT}$ (aa 1–350). The white circle marks the region of the phosphate-binding pocket. In the 90° rotated view, three arginines R108–110 are depicted in XErp1. (C) CSF extract was supplemented with IVT Flag-cyclin B3$^{WT}$ and the indicated Myc-tagged XErp1$^s$ variants. XErp1$^s$ is stabilized due to mutation of the two phospho-degrons (see Fig. EV2C). In XErp1$^{s\ 3RA}$, the three arginine residues C-terminal to T97 were replaced by alanine residues. Exit from MII was prevented by MG262 and IVT XErp1$^{CT}$. Following α-Flag IP, input, supernatant, and IP samples were immunoblotted. α-tubulin and p150 served as loading control. One representative experiment from three biological replicates is shown. Asterisk marks heavy chain of antibody used for IP. (D) CSF extract was supplemented with IVT Myc-XErp1* WT or 3RA and Flag-cyclin B3$^{WT}$. XErp1* is deficient in APC/C inhibition (see Fig. EV2C) to prevent it from interfering with meiotic exit. At indicated time points, samples were taken, treated with lambda phosphatase (λ) and immunoblotted. p150 served as loading control. One representative experiment from three independent replicates is shown. Source data are available online for this figure.

with mouse cyclin B3$^{WT}$ (Fig. 7C). In sharp contrast, mouse cyclin B3$^{PM}$ had no effect on the stability of Emi2* $^{WT}$. Furthermore, cyclin B3$^{WT}$ had no destabilizing effect on Emi2 when Emi2 carried a non-phosphorylatable mutation at position 86 (Emi2$^{T86A}$) or

when extract containing Emi2$^{WT}$ was co-treated with the Plk1 inhibitor BI2536 (Fig. 7C). From these experiments, we concluded that the pocket-dependent mechanism of XErp1/Emi2 degradation involving Plk1 is conserved from frog to mouse.

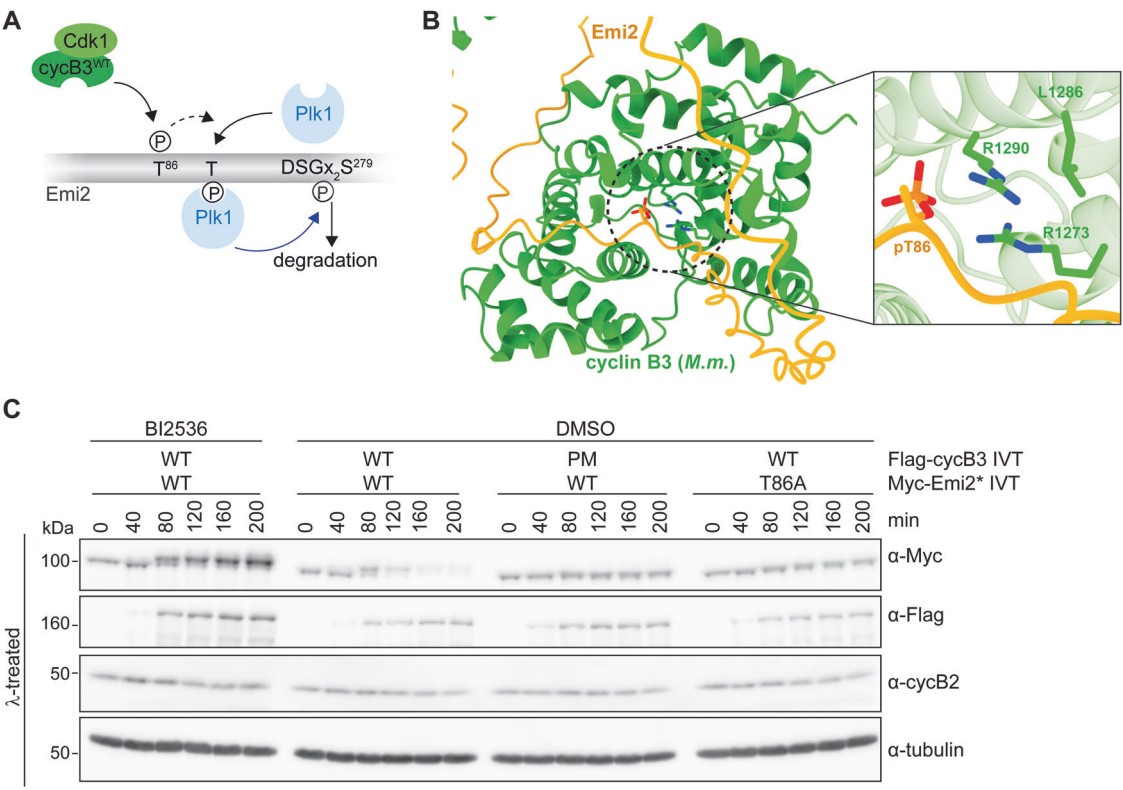

**Figure 7. The mechanism of cyclin B3-mediated degradation of XErp1/Emi2 is evolutionarily conserved.**

(A) Scheme of mouse (*M.m.*) cyclin B3-mediated degradation of Emi2. Note, that mouse Emi2 has only one phosphodegron (DSGx$_2$S$^{279}$). (B) AlphaFold-Multimer prediction of mouse (*M.m.*) cyclin B3 (aa 979–1397) with *M.m.* Emi2 (aa 1–342) was used to model phosphorylated T86 near the pocket. The residues R1273, L1286, and R1290 were mutated to T1273, E1286 and M1290, respectively, to create pocket mutant (PM) mouse cyclin B3. (C) CSF extract was supplemented with mRNA encoding Flag-tagged *M.m.* cyclin B3$^{WT/PM}$, IVT of WT or T86A *M.m.* Myc-Emi2 and BI2536 or the solvent control DMSO. At indicated time points, samples were taken, treated with lambda phosphatase (λ) and immunoblotted. α-tubulin served as loading control. One representative experiment from three biological replicates is shown. Source data are available online for this figure.

# Discussion

Mutations in Ccnb3 in human oocytes are associated with recurrent triploid pregnancies and miscarriage (Fatemi et al, 2021; Rezaei et al, 2022; Wang et al, 2023). Mice lacking cyclin B3 are viable with fertile males, but infertile females due to a precocious arrest of oocytes at metaphase of the first meiotic division. Recently, we dissected the molecular cause for these observations by demonstrating that the primary function of cyclin B3 during female meiosis I is to target the APC/C inhibitor XErp1/Emi2 for degradation (Bouftas et al, 2022). By constantly keeping the levels of XErp1/Emi2 below the threshold critical for APC/C inhibition as oocytes progress through meiosis I, cyclin B3 prevents an untimely CSF arrest already at metaphase I. Upon fertilization, such an arrest would have deleterious effects on embryo development. In fact, previous studies have shown that loss of cyclin B3 in mice can result in triploid embryos upon fertilization or intracytoplasmic sperm injection (Chotiner et al, 2022; Li et al, 2019b). Degradation of cyclin B3 at the exit from meiosis I clears the way for XErp1/Emi2 accumulation, and this is essential to drive oocytes into the second meiotic division without replicating their DNA. In this study, we dissected the molecular mechanism of Cdk1/cyclin B3-mediated degradation of XErp1. Notably, this mechanism involves a hitherto unknown phosphate-binding pocket in cyclin B3. Phosphopeptide pulldown and co-IP experiments revealed that the pocket binds XErp1

T97 (T86 in mouse Emi2) only when it is phosphorylated (Fig. 4B,C) and this binding event is essential for further phosphorylations (Fig. 5A,C), which ultimately target XErp1 for degradation. These observations raise two questions. First, which is the kinase that primes XErp1 by phosphorylating T97 and, second, what are the downstream phosphorylation events important for XErp1/Emi2 degradation. In line with our previous study (Bouftas et al, 2022), we demonstrate that T97, which is part of a highly conserved Cdk1 (S/T)P consensus motif (Fig. 3C), is directly phosphorylated by Cdk1/cyclin B3 and this phosphorylation does not depend on pocket integrity (Fig. 3D), i.e., Cdk1/cyclin B3 self-primes XErp1 at T97 in a pocket-independent manner. A similar mechanism of pocket-independent substrate phosphorylation and subsequent binding of the phospho-site by cyclins has also been observed for separase and Cdk1/cyclin B1 (Gorr et al, 2005; Yu et al, 2021). The phosphorylation events downstream of pT97 are more complex as they involve Plk1 in addition to Cdk1/cyclin B3. Confirming our previous study (Bouftas et al, 2022), we could demonstrate that XErp1 degradation during MI requires phosphorylation of T170 (Fig. EV3B) and that the recruitment of Plk1 to XErp1 is triggered by Cdk1/cyclin B3$^{WT}$ (Fig. 5C). While it is tempting to speculate that Cdk1/cyclin B3 recruits Plk1 to XErp1 by directly phosphorylating T170, the current evidence argues against this idea. First, if Cdk1/cyclin B3 bound to pT97 would directly phosphorylate T170, then T97 and T170 should be sufficient to destabilize XErp1 via Plk1-mediated

phosphorylation of the two phosphodegrons. However, as shown in Fig. 5B, this is not the case as XErp1[7A], which is wildtype for T97, T170, and the two phosphodegrons, is stable in the presence of cyclin B3[WT] demonstrating that among the known phosphorylation sites T97 is required, but not sufficient to cause XErp1 degradation. In fact, T170 neither matches the Cdk1 (S/T)P consensus motif nor does it fulfill the requirement of basic residues C-terminal to non-consensus Cdk1 phosphorylation sites (Suzuki et al, 2015). Importantly, a previous study concluded that Plk1 itself phosphorylates T170 (Isoda et al, 2011). This study investigating how the activity of Cdk1/cyclin B1 remains constant during the CSF arrest despite ongoing cyclin B1 synthesis revealed that Cdk1/cyclin B1 transiently destabilizes XErp1 resulting in temporary cyclin B1 destruction. According to this study, T170 phosphorylation by Plk1 involves a two-step recruiting mechanism with initial phosphorylation of XErp1 by Cdk1/cyclin B1 at multiple (S/T)P sites creating low-affinity binding sites for Plk1. Notably, Plk1 binds to these phosphosites independently of its phosphate-binding site, the polo-box domain (PBD) (Elia et al, 2003). Then, Plk1 phosphorylates T170 and thereby, creates its own high-affinity PBD-dependent docking site. Based on these and our data, we propose the following model: (1) Cdk1/cyclin B3 phosphorylates XErp1 T97. (2) Phosphorylated T97 recruits Cdk1/cyclin B3 via cyclin B3's phosphate-binding pocket. (3) Recruited Cdk1/cyclin B3 phosphorylates XErp1 at additional Cdk1 sites resulting in low affinity binding of Plk1. (4) Plk1 phosphorylates T170, binds to pT170 and phosphorylates the two phospho-degrons (DSGX$_3$S[38], DSAX$_2$S[288]) resulting in efficient XErp1 degradation via the ubiquitin ligase SCF[βTRCP]. Our data suggest (Fig. 7C) that this mechanism is conserved in frog and mouse.

Regarding the stability of XErp1/Emi2, meiosis I and meiosis II differ significantly. During MI, XErp1/Emi2 has to be degraded as efficiently as possible to prevent an unwanted, precocious CSF arrest in MI, which would have deleterious effects on embryo development upon fertilization. Consistently, defects in this pathway are associated with recurrent miscarriages in women (Fatemi et al, 2021; Rezaei et al, 2022; Wang et al, 2023). During MII, XErp1/Emi2 must be degraded on the one hand as much as necessary to ensure that Cdk1/cyclin B1 activity does not overshoot due to continuous cyclin B1 synthesis and on the other as little as possible to not endanger the robustness of the CSF arrest. Notably, both cyclin B1 and cyclin B3 are expressed in meiosis I, while only cyclin B1 is present in meiosis II. Since both cyclins possess a functional phosphate-binding site, the question arises why Cdk1/cyclin B1 is not efficient in targeting XErp1/Emi2 for degradation? As we have shown previously (Bouftas et al, 2022), XErp1 T97 is only a poor substrate for Cdk1/cyclin B1, while it is very efficiently phosphorylated by Cdk1/cyclin B3 and this phosphorylation occurs in a phosphate-binding pocket-independent manner (Fig. 3D). Given the importance of T97 phosphorylation for all subsequent events, which then culminate in XErp1/Emi2 degradation, differences in T97 substrate preference between Cdk1/cyclin B1 and Cdk1/cyclin B3 would be sufficient to explain differential XErp1/Emi2 stability in MI and MII. However, our data suggest that this is not the whole story as cyclin B1 and cyclin B3 seem to differ also in their ability to bind phosphorylated T97 via their respective phosphate-binding pocket (Fig. 6A). Differences in residue composition, and, hence, in the surface charge in proximity to the pocket seem to contribute to different substrate specificity of cyclin B3 and B1. Motivated by the AlphaFold2 prediction, we experimentally tested if the cluster of three arginine residues C-terminal to XErp1 T97 (R108, R109, R110) (Fig. 3C) contributes to the interaction with cyclin B3, which possesses negatively charged residues in proximity of its pocket (Fig. 6B).

Indeed, mutating the arginine residues to alanines prevented association of XErp1 with cyclin B3 (Fig. 6C) and, consequentially, resulted in a significant stabilization of XErp1 (Fig. 6D). Thus, differences in residues forming and surrounding the phosphate-binding pocket in cyclin B3 and B1 seem to contribute to substrate specificity. As mentioned above, a conserved histidine residue (H320) in cyclin B1 is replaced by leucine in cyclin B3. Our phosphopeptide analyses suggest that cyclin B3's pocket preferentially binds phosphorylated threonine residues, rather than phosphorylated serine residues (Fig. EV3A). Further studies are required to investigate if the preference for pThr is a generic characteristic of cyclin B3, or only valid in the context of XErp1. Of note, cyclin B1 binds via its phosphate-binding pocket to phosphorylated serine 1126 of separase (Yu et al, 2021) and we could demonstrate in a study parallel to this one that cyclin B1's phosphate-binding pocket can bind not only phosphorylated serine, but also phosphorylated threonine or even acidic residues (Heinzle et al, 2024). The cause for the differential binding preference of cyclin B1's and cyclin B3's phosphate-binding pocket merits further investigations.

The phosphate-binding pocket in cyclin B1 was identified in a study elucidating the structure of human separase bound to Cdk1/cyclin B1 (Yu et al, 2021). Both pocket integrity and phosphorylation of separase at S1126 are critical for complex formation and, thus, for mutual inhibition of both enzymes (Stemmann et al, 2001; Yu et al, 2021). Key residues forming a hydrogen-bonding network with the phosphate attached to S1126 are highly conserved within B-type cyclins, but not in any other cyclin family. Notably, despite the high degree of sequence conservation across species including yeast, the mechanism of mutual inhibition of separase and Cdk1/cyclin B1 is restricted to vertebrates (Hellmuth et al, 2015), suggesting that the pocket evolved originally to fulfill separase-independent functions. Motivated by this idea, we searched for Cdk1/cyclin B1 substrates that are sequentially phosphorylated in a pocket-dependent manner. This search led to the identification of the APC/C, whose full activation depends on pocket-dependent phosphorylation events (Heinzle et al, 2024). Similarly, for yeast it was recently reported that multisite phosphorylation of the transcriptional co-activator Ndd1 by Cdk1 seems to involve both Cks1 and the phosphate-binding pocket of the B-type cyclin Clb2 (Asfaha et al, 2022). These observations raise the question if XErp1/Emi2 is the only substrate of Cdk1/cyclin B3 that is phosphorylated in a phosphate-binding pocket-dependent manner. Indeed, our IP experiment followed by WB analyses using a pan-specific phospho-threonine or phospho-serine antibody suggests that there are more substrates than XErp1/Emi2 (Figs. 1D and EV1B). Thus, further studies are required to understand the full complexity of pocket-dependent substrate phosphorylation by Cdk1/cyclin B3.

## Methods

### Reagents and tools table

| Reagent/resource | Reference or source | Identifier or catalog number |
| --- | --- | --- |
| **Experimental models** | | |
| *Xenopus laevis* adult female frogs | bred in-house | |
| **Recombinant DNA** | Accession numbers | |
| For IVT and mRNA pCS2 plasmid were used | | |
| For expression in Sf9 cells pMACS was used, for expression in *E. coli* pMal plasmids were used | | |

| Reagent/resource | Reference or source | Identifier or catalog number |
|---|---|---|
| Xl_cyclin B1 | NM_001088520.1 | |
| Xl_cycB3 | NM_001085892.2 | |
| Xl_XErp1 | NM_001099868.1 | |
| Mm_cyclin B3 | NM_183015.4 | |
| Mm_Emi2 | NM_001081253.2 | |
| **Antibodies** | | |
| Rabbit anti-Cks2 | Self-made against CLFRRPLP KEQQK and CLFRRPLPKDQQK peptide | This study |
| Mouse anti-cyclin B2 | Santa Cruz | sc-53239 |
| Rabbit anti-cyclin B3 | Self-made (purified against n-terminal CycB3 aa 1-110) | Bouftas et al, 2022 |
| Rabbit anti-cyclin B3 | Self-made (against aa 6-22: RPSRPVASKLPKLGKPVC peptide) | Bouftas et al, 2022 |
| Mouse anti-Cdk1 | Santa Cruz | Sc-54 |
| Mouse anit-ppCdk1 | Gift from Tim Hunt | |
| Rabbit anti-APC3 | Self-made | Schmidt et al, 2005 |
| Mouse anti-Flag | Sigma-Aldrich | F1804 |
| Mouse anti-tubulin | Monoclonal antibody clone AA4.3 | |
| Mouse FITC-anit-tubulin | Sigma | F2168 |
| Rabbit anti-Plk1 | Self-made | |
| Mouse anti-p150 | BD Transduction Laboratories | 610473 |
| Rabbit anti-XErp1 | Self-made | Schmidt et al, 2005 |
| Rabbit anti-pT97 XErp1 | Self-made against pT97 peptide: CQLAPLYE(p)TPRVGKKE | This study |
| Mouse anti-pThr | Cell Signaling | #9386 |
| Mouse anti-pSer | Santa Cruz | sc-81514 |
| Rabbit anti-Separase | Gift from Olaf Stemmann | |
| Streptavidin-HRP | ThermoScientific | #21130 (LotNumber RK240385) |
| Goat Secondary anti-mouse for western blot | Jackson Immuno-research | 115-035-146 |
| Goat Secondary anti-rabbit for western blot | Jackson Immuno-research | 111-035-144 |
| **Oligonucleotides and other sequence-based reagents** | | |
| Primers | Bouftas et al, 2022 | |
| Primers | This study: | |
| Xl_CycB1.S fwd | attaGGCCGGCCaATGGCTCTGATTATGACAAGGAGT | |
| Xl_CycB1.S rev | taatGGCGCGCCcCTATGATGCAGCTTTTGCCAG | |
| Xl_CycB3_Pocket I for | ACCATGGAGACTGAGACCCTTGCCATGTACATTTGTGAA | |
| Xl_CycB3_Pocket I rev | TTCACAAATGTACATGGCAAGGGTCTCAGTCTCCATGGT | |
| Xl_CycB3_PocketI I for | TACCGCTTCCTGACCAGATTTGCTAAG | |
| Xl_CycB3_PocketI I rev | CTTAGCAAATCTGGTCAGGAAGCGGTA | |
| Xl_XErp1 fw | attaggccggccAatggcTaaCctcttagagaatttttgctgccc | |
| Xl_XErp1-Aa350 wo stop codon | atatGGCGCGCCAtccttcttcctctacttc | |
| Xl_XErp1 91-126aa | CCAATGCTTGCTCCACTTTATGAAACCCCTCGAGTTGGGAAGAAGGAATTCTCACTTCGCAGAAGGCTGCTCATTTCTAAAGCTACCTCGGGAGGAAATTTAGATTTTGATGTGAGG | |
| XL_Cyclin B3"short"_rev | taatggcgcgccTTAGCTCTGAAGGGCCTCTGTA | |

| Reagent/resource | Reference or source | Identifier or catalog number |
|---|---|---|
| Xl_CyclinB3_for_ AscI cut for insert in XErp1 | GGATGGcgcgccATGATGCCTTCTCTTCGTCCATCTC | |
| Xl_XErp1_R108-110A_fw | GAATTCTCACTTgcCGcAgcGCTGCTCATTTC | |
| Xl_XErp1_R108-110A_rev | GAAATGAGCAGCgcTgcGgcAAGTGAGAATTC | |
| **Chemicals, enzymes and other reagents** | | |
| pT97-peptide | CQLAPLYE(p)TPRVGKKE | PSL |
| T97-peptide | CQLAPLYETPRVGKKE | PSL |
| pS97-peptide | CQLAPLYE(p)SPRVGKKE | PSL |
| S97-peptide | CQLAPLYESPRVGKKE | PSL |
| pT170-peptide | PIATSTLKTESESGTC | PSL |
| T170-peptide | | PSL |
| BI2536 | Final concentration: 20 µM, ICS | #S1109 |
| Cytochalasin B | Final concentration: 0.01 mg/ml, Cayman Chemical/ Biomol | Cay11328 |
| PhosStop | Roche/Sigma | #4906837001 |
| cOmplete Protease inhibitor cocktail | Sigma/Merck | #5056489001 |
| Flavopiridol | Final concentration: 10 µM Biozol/ sellekchem | #S1230 |
| Hoechst 33342 | Final concentration: 1 µg/ml ThermoFischer | H21492; CAS#: 23491- 52-3 |
| MG262 | Final concentration: 100 µM Sigma | #539163 |
| Okadaic acid | Final concentration: 1-10 µM Millipore/Sigma | #459620 |
| Progesterone | Final concentration: 5 µg/ml, Sigma | #P8783 |
| **Software** | | |
| AlphaFold | https://doi.org/10.1101/2021.10.04.463034 | Version 2.3.2 |
| ImageJ-Fiji | https://imagej.nih.gov/ij/ | Version 2.0 |
| GraphPad Prism 8 | https://www.graphpad.com/scientific-software/prism | Version 8.4.0 |
| GelAnalyzer | http://www.gelanalyzer.com/?i=1 | Version 19.1 |
| **Other** | | |
| TNT® SP6 High-Yield Wheat Germ Protein Expression System | Promega | L3260 |
| mMESSAGE mMACHINE® T7 Ultra Kit | Ambion | AMB1345-5 |
| NucleoBond Xtra Midi kit | Macherey-Nagel | 740410.100 |
| QIAGEN gel extraction kit | QIAGEN | 28704 |
| QIAprep Spin Miniprep Kit | QIAGEN | 27106 |

## Purification of recombinant proteins

For purification of variants of cycB3/Cdk1$^{AFE}$ complexes, Sf9-cells were co-infected with TwinStrep-cycB3 variants and His-Cdk1 and were harvested 48–52 h post-infection by centrifugation at $1000 \times g$, 4 °C for 15 min. Cells were resuspended with lysis buffer (PBS, 0.5 M NaCl, and 1× Protease-Inhibitors (complete tablets w/o EDTA (Roche)). After freezing, the thawed pellet was sonicated with Branson sonicator (5 min w/o pulse). Lysate was cleared by

two centrifugations each for 15 min at $50,000 \times g$ and 4 °C. Triton-X-100 was added to the supernatant to a final concentration of 0.25%. After filtration (0.45 µm) Nuclease was added (1:10,000 Universal Nuclease Pierce final 25 E/ml) and the lysate was incubated for 4 h with StrepTactin Superflow (IBA) beads. Beads were washed six times with PBS/0.5 M NaCl (two times in batch and four times in column) and once with PBS. The cycB3/Cdk complex was eluted with PBS containing 10 mM desthiobiotin.

For purification of XErp1 1–350 aa fragments, XErp1 mutants were cloned into a pMALvector with N-terminal MBP tag and C-terminal His-tag as well as two TEV cleavage sites after the MBP-tag. Expression took place in BL21 DE3 RIL *E. coli* induced with 500 µM IPTG at an OD600 of 0.5 for 3–4 h at 37 °C. Bacteria were centrifuged for 10 min at $3000 \times g$ and the pellet was resuspended in 20 ml Lysis buffer (20 mM Tris-HCl pH 8.0, 300 mM NaCl, 5 Mm Imidazole and 1× Protease Inhibitor). After snap-frozen in liquid nitrogen cell pellets were stored at −80 °C until purification. For purification thawed bacteria pellet solution was lysed with the Emulsi-Flex-C3. Lysate was cleared by centrifugation at $20,000 \times g$, 4 °C for 30 min. The supernatant was mixed with 1 ml Ni-NTA beads (Qiagen) and washed with 10 ml PBS before use. Beads and lysate were incubated for 2 h at 4 °C rotating. Beads were washed three times with washing buffer (20 Mm Tris HCl pH 8, 300 mM NaCl, 20 mM Imidazole, 0.1% Triton-X-100) containing 1 mM ATP and 1 mM MgCl₂ and three times with washing buffer. Elution took place with elution buffer (20 mM Tris HCl pH 8.0, 300 mM NaCl and 200 mM Imidazole) in 10 fractions. Peak fractions were identified via Coomassie, pooled and dialysed with PBS containing 10% Glycerol o/n 4 °C. For storage dialysed protein was aliquoted, snap frozen and kept at −80 °C.

## Synthesis of mRNA

For mRNA synthesis, the plasmid containing the gene of interest (pCS2 backbone) was linearized with a restriction enzyme, either XbaI or AscI, for 5 to 6 h. In total 10 µg DNA was digested with 0.8 U/µl enzyme in 50 µl reaction volume. DNA was precipitated by the addition of 2.5 µl 0.5 M EDTA, 5 µl 5 M ammonium acetate and 100 µl 100% EtOH for at least 1 h at −20 °C.

Precipitated DNA was centrifuged for 30 min at full speed at 4 °C and the dry pellet was solved in water. For in vitro transcription, the mMESSAGE mMACHINE™ T7 ULTRA transcription kit (Invitrogen) was used. All incubation steps were performed at 37 °C and work was performed RNase-free. Transcription was performed with 1 µg linearized DNA supplemented with 10 µl NTP/ARCA buffer, 2 µl 10× T7 reaction buffer, 2 µl T7 enzyme mix, 1 µl RNAsin and filled up to 20 µl with water. The reaction mix was incubated for 2.5 h. For poly-A-tailing, first DNA was digested by the addition of 1 µl TURBO DNase for 15 min, then 36 µl water, 20 µl E-PAP buffer, 10 µl 25 mM MnCl₂, 10 µl ATP solution and 4 µl E-PAP was added. After incubation for 1 h mRNA was precipitated by the addition of 50 µl LiCl₂. For full precipitation, the mix was incubated overnight at −20 °C and centrifuged for 30 min at full speed at 4 °C on the following day. mRNA pellet was washed once with 200 µl 70% EtOH, dried at 37 °C for 10 to 15 min and solved in water. After adjusting mRNA concentration to 1–1.7 µg/µl, aliquots were snap frozen in liquid nitrogen and stored at −80 °C.

## In vitro transcription and translation

For the production of in vitro transcribed and translated proteins (IVT), TNT® SP6 High-Yield Wheat Germ Protein Expression System (Promega) was used. For a reaction volume of in total 50 µl, 10 µg plasmid containing the gene of interest (in a pCS2 vector) was mixed with 30 µl wheat germ extract and filled up with water (MiliQ) to 50 µl. The reaction mix was incubated for 2.5 h and afterwards aliquoted, snap-frozen in liquid nitrogen and stored at −80 °C.

## *Xenopus laevis* stage VI oocytes and CSF extract

*Xenopus laevis* frogs were bred and maintained under laboratory conditions at the animal research facility, University of Konstanz, and all procedures performed were approved by the Regional Commission, Freiburg, Germany. For removal of stage VI oocytes frogs were anesthetized in Tricaine solution (1 g in 1 L of MMR (5 mM Na-HEPES, 0.1 mM EDTA, 0.1 mM NaCl, 2 mM KCl, 1 mM MgCl₂, 2 mM CaCl₂; pH 7.8). Ovaries were surgically removed according to approved proposal (file number: 35-9185.81/G-22/080) and maintained in MBS buffer (88 mM NaCl, 10 mM KCl, 10 mM MgSO₄, 25 mM NaHCO₃, 0.7 mM CaCl₂, 50 mM HEPES pH 7.8). For CSF preparation, frogs were primed with injection of 20 U human chorionic gonadotropin (hCG) 3 to 14 days previous to injection of 500 U hCG to induce egg laying according to approved proposal (file number: 35-9185.81/G-22/081). Further CSF preparation was performed by step wise centrifugation of dejelled eggs (described in detail in Bouftas et al, 2022).

## Immunoprecipitation (IP) experiments

For IP experiments, antibody was bound to Dynabeads™ Protein G (Invitrogen) according to producer protocol. Depending on the CSF extract or IVT amount 1 µg antibody per 10 µl CSF extract or 4 µg antibody per 2 µl IVT was used for binding. Previous to IP, beads were washed 3x with PBST and 1× with PBS.

For the comparison of binding partners of cycB1 and cycB3, CSF extract was supplemented with 100 µM MG262 and 10 µM okadaic acid to maintain the extract in metaphase II with an active APC/C. CSF was distributed and either control IVT or Flag-tag cycB1 or cycB3 was added (1:10–1:20) and extract was incubated at 20 °C for 15 min. IP was performed with 60 µl extract for 1 h at RT on a spinning wheel.

Flag-IP of Flag-cycB3$^{WT}$ or Flag-cycB3$^{PM}$ was performed in CSF extract arrested with 1:20 Myc-XErp1$^{CT}$ (aa 491–651, T545A, T551A) and MG262 (100 µM) in metaphase II. IVT of Flag-cycB3 (1:20–1:24) and Myc-XErp1 variants (1:20) were added and incubated for 10 min at 20 °C. IP was performed for 30 min at RT. Afterwards beads were treated with one half of IP volume λ-phosphatase mix and incubated for 50 min at 30 °C. To stop λ-phosphatase reaction the other half of IP volume of 3× sample buffer was added.

Flag-IP of Flag-XErp1$^s$ was performed in CSF extract arrested with 1:20 Myc-XErp1$^{CT}$ (aa 491-651, T545A, T551A) and MG262 (100 µM) in metaphase II. IVT of Myc-cycB3 (1:10) and Flag-XErp1$^s$ (1:20) constructs were incubated for 30 min at 20 °C. Previous to IP, CSF extract was diluted 1:4 with IP buffer (1× Protease Inhibitor, 1× PhosStop, 20 mM Tris-HCl, pH 7.5, 100 mM NaCl, 10 mM EDTA, 5 mM NaF, 1 mM Na₃VO₄, 1 mM DTT). IP was performed for 30 min at RT.

After IP, beads were washed three times with PBS with 0.025% Tween 20 or two times with PBST and once with PBS. Samples were taken at indicated time points and diluted in 1.5× sample buffer (90 mM Tris HCl pH 6.8, 5% SDS (w/v), 15% glycerol, 7.5% β-Mercaptoethanol (w/v), Bromphenol blue).

## Coupling of (p)T97 (XErp1 90–104) peptides to SulfoLink™ coupling resin and peptide pulldown experiment

SulfoLink beads (Thermo Fisher Scientific) were washed with coupling buffer (50 mM Tris, 5 mM EDTA, pH 8.5) and incubated with 4 mg peptide per 1 ml beads. XErp1 90–104 aa peptide (PSL) either phosphorylated at T97 or non-phosphorylated (aa: CQLA-PLYE(p)TPRVGKKE) were dissolved in PBS. Incubation took place at RT for 30 min on a spinning wheel. After settling the beads for additional 30 min, beads were washed with coupling buffer, and incubated with quenching buffer (50 mM L-cystein in coupling buffer) for 15 min on a spinning wheel and additional 30 min upright. After quenching, beads were washed 3× with 1 M NaCl and 3× with PBS. Peptide concentrations in input and supernatant were compared for coupling efficiency by measurement of 280 nm extinction coefficients with a NanoDrop Spectrophotometer. Beads were stored in PBS containing 1× cOmplete protease inhibitor cocktail and 1× PhosSTOP™ at 4 °C.

In CSF extract arrested in metaphase II with proteasome inhibitor MG262 (100 µM) and Myc-XErp1$^{CT}$ IVT (1:20), mRNA of Flag-cyclin variants was added to the extract at a final concentration of 0.012 µg/µl and incubated at 20 °C for 50 min. After incubation, the extract was diluted 1:5 in pre-chilled IPB (20 mM Tris-HCL, pH 7.5, 100 mM NaCl, 10 mM EDTA, 5 mM NaF, 1 mM Na$_3$VO$_4$, 1 mM DTT). 5 volumes of diluted CSF were then added to 1 volume peptide-coupled SulfoLink beads (10–30 µl) and incubated for 30 min at 4 °C on a spinning wheel. After incubation, beads were washed two times with IPB buffer with 0.1% tween40 and two times with IPB buffer. Bead samples were then obtained by adding 1.25 volumes of 1.5× sample buffer to 1 volume beads, heating at 95 °C for 5 min, and centrifugation at 900 × g for 1 min, followed by the collection of the supernatant.

## Radioactive kinase assay

Radioactive kinase assays were performed to test activity of purified cycB3/Cdk1 complexes against either histone H1 or XErp1 1-350 aa constructs. Either cycB3$^{WT}$ or cycB3$^{PM}$ was co-purified with the kinase Cdk1 and used for the kinase assay. Around 8 to 17 ng/µl cycB3/Cdk1 were incubated with 40 ng/µl Histone H1 or 80 ng/µl XErp1, 50 µM ATP and 0.22 µCi/µl [γ-$^{33}$P]ATP in PB-buffer (PBS with 1% Glycerol, 20 mM EGTA, 15 mM MgCl$_2$, 1 mM DTT, 1× Protease inhibitor (complete w/o EDTA), 1× PhosStop (Roche)). The reaction took place at 25 °C and 1400 rpm shaking. Samples were taken at indicated time points and directly diluted in 3× sample buffer to a final concentration of 1×.

## Oocyte TRIM away

Ovaries from frogs were manually dissected in smaller pieces and treated for 1.5 h with Liberase (50 µg/ml) in MBS buffer. After washing separated oocytes, stage VI oocytes were collected. Stage VI oocytes

were injected with codon-optimized Flag-Trim21 mRNA (7 ng/oocyte), cycB3 antibody (against aa 6-22 RPSRPVASKLPKLGKPVC peptide, Bouftas et al, 2022) or control IgG (29 ng/oocyte) and IVT of cycB3$^{WTΔ15}$, cycB3$^{PMΔ15}$ or control IVT respectively (1.5 nl to 6.55 nl/oocyte). In total a volume of 18.4 nl was injected per oocyte. After incubation for 14–16 h at 19 °C in MBS-buffer, oocytes were incubated with OR2 medium (82.5 mM NaCl, 2.5 mM KCl, 1 mM CaCl$_2$, 1 mM MgCl$_2$, 1 mM Na$_2$HPO$_4$, 5 mM HEPES pH 7.8) containing progesterone (5 µg/ml) and samples were taken at indicated time points by removing buffer and snap-frozen in liquid nitrogen.

For immunofluorescence staining oocytes were fixed at 4 h after GVBD in 100 mM KCl, 3 mM MgCl$_2$, 10 mM K-HEPES pH 7.8, 3.7% Formaldehyde, 0.1% Glutaraldehyde and 0.1% Triton X-100, overnight at 4 °C shaking. Oocytes were bleached in 10 °% H$_2$O$_2$ in MeOH at RT exposed to light for one day. After bleaching oocytes were blocked three times with AbDil (2% BSA, 0.1% NaN$_3$ and 0.1 °% Triton X-100 in PBS) for 1 h at 4 °C on a shaker. For antibody staining, FITC-labelled anti-tubulin antibody was diluted to a final concentration of 1 µg/ml and for DNA staining Hoechst 33342 was added to a final concentration of 1 µg/ml in AbDil. Oocytes were then incubated for at least 48 h to 72 h at 4 °C in the dark on a shaker. After staining oocytes were washed three to four times with PBST. Mounting took place on slides with holes in a diameter of 1.2 µm–1.3 µm with the vacuum grease ring method.

## XErp1 degradation assay in CSF

For investigation of degradation behavior of different XErp1 mutants, CSF extract was incubated with 1:20 to 1:40 µl Myc-XErp IVT for 5–10 min at 20 °C. The time course was started by addition of Flag-cycB3 IVT (1:13 to 1:20). At the respective time points samples were taken and either directly diluted 1:10 in 1.5× sample buffer or added to the same amount of λ-Phosphatase mix and dephosphorylated for 30–40 min at 30 °C. For comparison of cycB3$^{WT}$ and cycB3$^{PM}$, no ectopic XErp1 construct was supplemented.

For testing mouse Emi2 degradation with mouse cyclin B3, Flag-cyclin B3 mRNA (5 µg/40 µl CSF extract) was supplemented to CSF extract containing Myc-Emi2 IVT variants (1:80). Samples were taken at indicated time points and directly added to λ-Phosphatase mix and dephosphorylated.

## λ-Phosphatase treatment

For the dephosphorylation of CSF extract samples, two phosphatase mixes were used depending on whether the sample was loaded with non-λ-phosphatase-treated samples or not. If only CSF samples were compared via SDS-PAGE and immunoblot, CSF extract was mixed 1:2 with λ-phosphatase mix, containing 1 volume of λ-Phosphatase (NEB) with 1 volume of water, 10 volumes 10 mM MnCl$_2$ and 10 volumes of 10× PMP buffer (NEB). The phosphatase mix for comparing untreated with treated samples contained 1:20 λ-phosphatase in 1 mM MnCl$_2$ and 1× PMP-buffer, CSF extract was diluted 1:5 with the phosphatase mix. Dephosphorylation took place for 30 to 40 min at 30 °C.

For dephosphorylation of oocyte lysate, oocytes were lysed without NaF and β-Glycerophosphate and incubated with 1:4 λ-Phosphatase mix, containing 1 volume λ-Phosphatase with 2 volumes 10× PMP buffer and 2 volumes 10 mM MnCl$_2$, after lysis. Samples were incubated for 30–45 min at 30 °C. The

dephosphorylation reaction was stopped by the addition of sample buffer, either 1:5 or 1:2 for CSF extract or 1:3 for oocyte lysate.

## IP for Cks2 depletion from CSF extract

For Cks2 depletion from CSF extract, 6 µg of Cks2 antibody for 70 µl CSF extract were bound to beads. Depletion took place over three rounds of Cks2 pull-down for 30 min at RT. After depletion, the supernatant extract was used for time course experiments.

## Western blot (WB)

For western blot analysis, protein samples were separated via SDS-PAGE and transferred to Nitrocellulose blotting membrane Protran 0.45 µm (Fisher Scientific) via wet blot. Detection of respective proteins was performed with following antibodies: α-Xl_Cks2 (self-made against CLFRRPLPKEQQK and CLFRRPLPKDQQK peptide, rabbit, 1 µg/ml), α-Xl_cycB2 (Santa Cruz sc-53239, mouse, 0.4 µg/ml), α-Xl_cycB3 (self-made against N-terminal cycB3 aa 1–110, rabbit, 0.15 ng/ml), α-Hs_Cdk1 (Santa Cruz sc-54, mouse, 1 µg/ml), α-Hs_ppCdk1 (gift from Tim Hunt, mouse, 1:1000), α−Xl_APC3 (self-made, rabbit, 0.5 µg/ml, Bouftas et al, 2022, Heim et al, 2015), α-Flag (Sigma-Aldrich F1804, 1 µg/ml), α-tubulin (monoclonal antibody clone AA4.3, 1:100), α-Xl_Plk1 (self-made, rabbit, 0.12 ng/ml), α-p150 (BD Transduction Laboratories (610473), mouse, 0.25 µg/ml), α-Xl_XErp1 (self-made against aa 1–300, rabbit, 0.53 µg/ml), α-pThr (cell signaling, #9386), α-pSer (Santa Cruz, sc-81514), α-Xl_Separase (gift from Olaf Stemmann). For detection of primary antibody HRP-coupled α-rabbit or α-mouse (Jackson Immuno Research Laboratories, goat, 1:5000–1:10,000) or α-rabbit (confirmation specific, cell signaling technology, mouse, 1:5000) were used. For detection of the secondary antibody, membrane was laid in sufficient SuperSignal West Pico PLUS (Thermo Fisher) solution and then imaged on the LAS-3000 (Fujifilm).

## AlphaFold2 Model

Multimer prediction of full-length cyclin B3 (*Xenopus laevis*) and XErp1 1–350 aa were performed as described in Evans et al, 2022. The highest ranked model was visualized and analysed in ChimeraX 1.6.1. For *M.m.* cyclin B3 and Emi2 multimer prediction was performed with *M.m.* cyclin B3 aa 979–1397 and Emi2 aa 1–342. The phosphorylated threonine was inserted with Coot (Crystallographic Object-Oriented Toolkit) using distance restraints to avoid steric clashes.

## Plasmids

In all experiments, the source of IVT or mRNA was pCS2 plasmids containing a T7 and SP6 promoter upstream of the gene of interest and the respective tags described in each experiment. Accession number of genes used: Xl_cycB1 NM_001088520.1; Xl_cycB3 NM_001085892.2; XErp1 NM_001099868.1; Mm_cycB3 NM_183015.4; Mm_Emi2 NM_001081253.2

# Data availability

This study includes no data deposited in external repositories.

The source data of this paper are collected in the following database record: biostudies:S-SCDT-10_1038-S44319-024-00347-8.

# Peer review information

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

## Acknowledgements

This work was supported by funding of the German Research Foundation (DFG, TUM: 456916924 and CRC969, B01: AH and TUM). We thank K. Diederichs and the Bioimaging Facility for support regarding AlphaFold predictions and oocyte imaging, respectively.

## Author contributions

**Rebecca Schunk**: Conceptualization; Investigation; Visualization; Methodology; Writing—review and editing. **Marc Halder**: Investigation. **Michael Schäfer**: Investigation. **Elijah Johannes**: Investigation. **Andreas Heim**: Conceptualization; Project administration; Writing—review and editing. **Andreas Boland**: Software; Writing—review and editing. **Thomas U Mayer**: Conceptualization; Funding acquisition; Methodology; Writing—original draft; Project administration; Writing—review and editing.

Source data underlying figure panels in this paper may have individual authorship assigned. Where available, figure panel/source data authorship is listed in the following database record: biostudies:S-SCDT-10_1038-S44319-024-00347-8.

## Funding

## Disclosure and competing interests statement

The authors declare no competing interests.

# Expanded View Figures

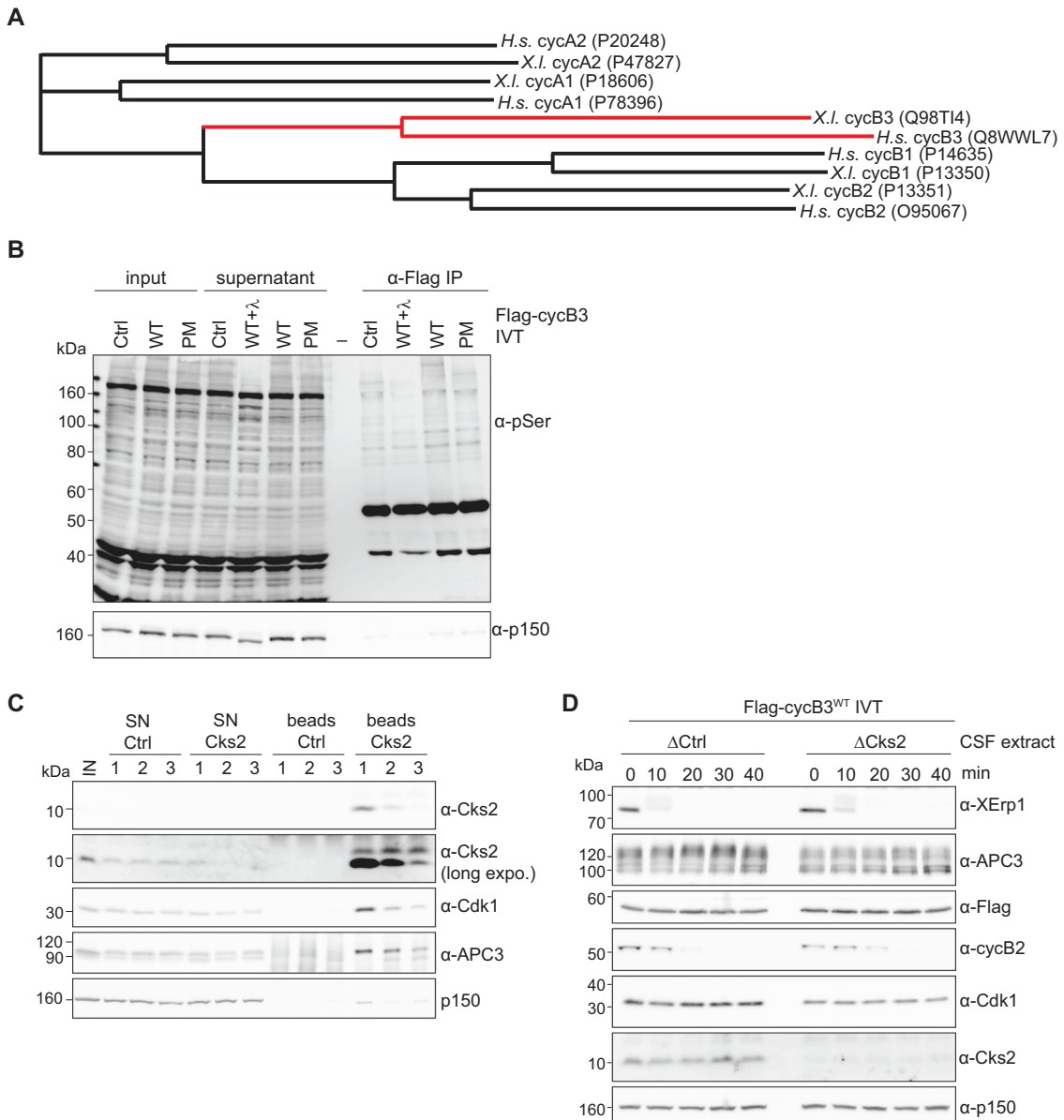

**Figure EV1. Phylogenetic tree of A- and B-type cyclins, pSer signal of cycB3^WT/PM IP and XErp1 degradation in Cks2 depleted CSF extract.**

(A) Phylogenetic tree of human (*H.s.*) and Xenopus (*X.l.*) cyclin (cyc) A1, cyclin A2, and B-type cyclins (B1-3) created within UniProt. In brackets, UniProt number of respective proteins are indicated. (B) pSer WB analysis of α-Flag IP samples shown in Fig. 1D. CSF extract was supplemented with IVT Flag-cyclin B3^WT/PM and the meiotic state of the extract was maintained by the addition of MG262 and okadaic acid. Indicated samples were treated with λ-phosphatase (+ λ). p150 served as a loading control. One representative experiment from three independent biological replicates is shown. (C) WB analysis of Cks2 depletion experiment. Cks2 was depleted from CSF extract by three rounds of α-Cks2 immunoprecipitations. Control (Ctrl) depletion was performed using unspecific IgG antibodies. Shown are input (IN), supernatant (SN) and bead samples. p150 served as loading control. (D) Control depleted or Cks2 depleted CSF extract was supplemented with IVT of Flag-cyclin B3^WT. Samples were taken at indicated time points and immunoblotted. One representative experiment from three biological replicates is shown. p150 served as loading control.

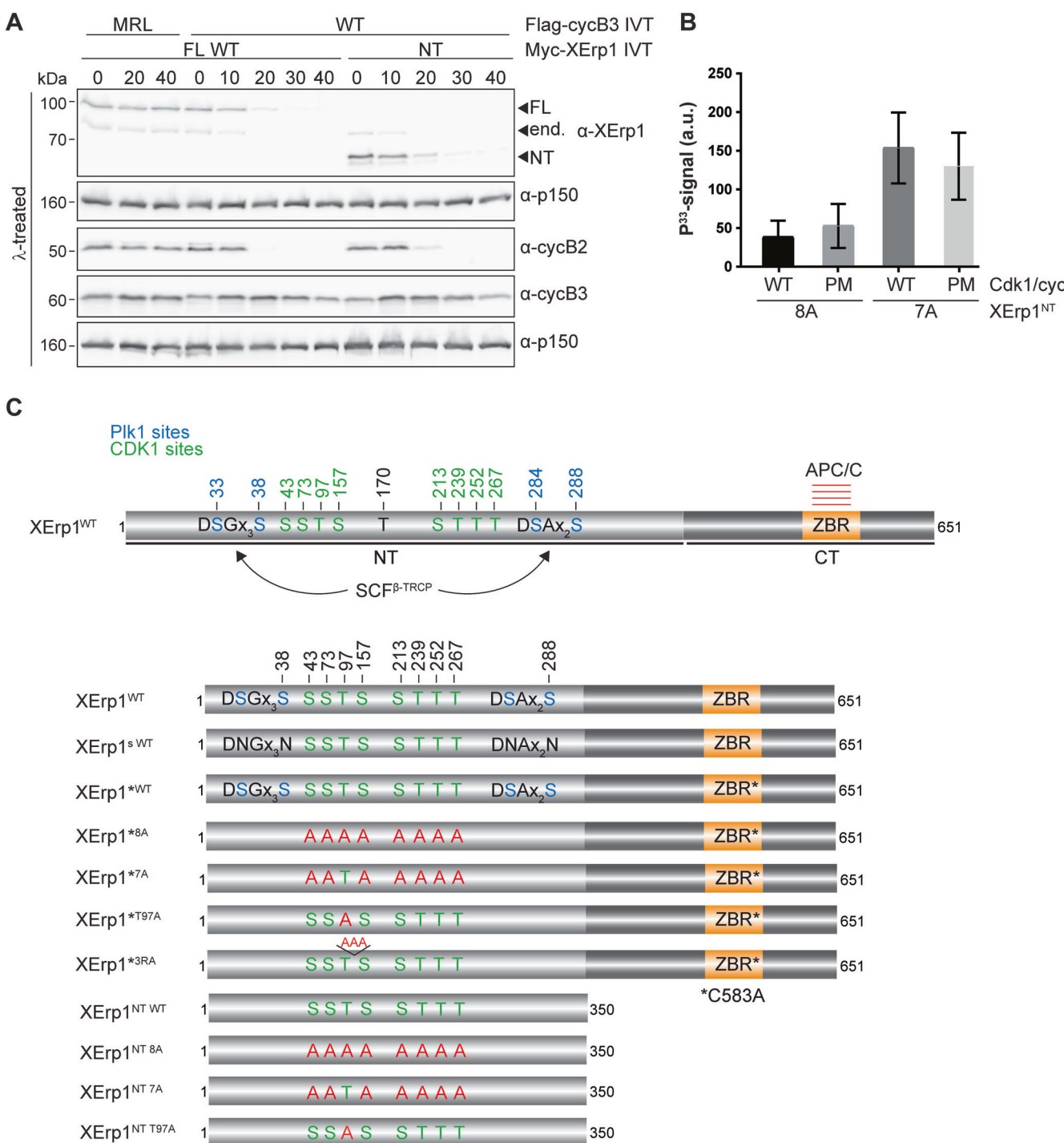

**Figure EV2. Degradation behaviour of XErp1 N-terminal fragment, quantification of T97 phosphorylation by Cdk1/cycB3$^{WT/PM}$ and schemes of XErp1 variants.**

(A) CSF extract was supplemented with Flag-cyclin B3$^{WT/MRL}$ and full-length (FL) or NT Myc-XErp1. Samples were taken at indicated time points and immunoblotted. p150 served as loading control. One representative experiment from two biological replicates is shown. (B) Quantification of experiment shown in Fig. 3D and two further replicates. The $^{33}$P signal was quantified and normalized to Cdk1 levels detected by WB. Mean ± SD is depicted ($n = 3$). (C) Illustration of XErp1 variants used for in vitro kinase assays (Figs. 3D and 5A) and experiments shown in Figs. 3D, 4C, 5A–C and 6C, D. XErp1$^s$ is a stable variant due to the mutation of both phosphodegrons (DSGx$_3$S$^{38}$→DNGx$_3$N$^{38}$ and DSAx$_2$S$^{288}$→DNAx$_2$N$^{288}$). XErp1* is deficient in APC/C inhibition due to a mutation in the zinc-binding region (ZBR, C583A). NT N-terminus, CT C-terminus.

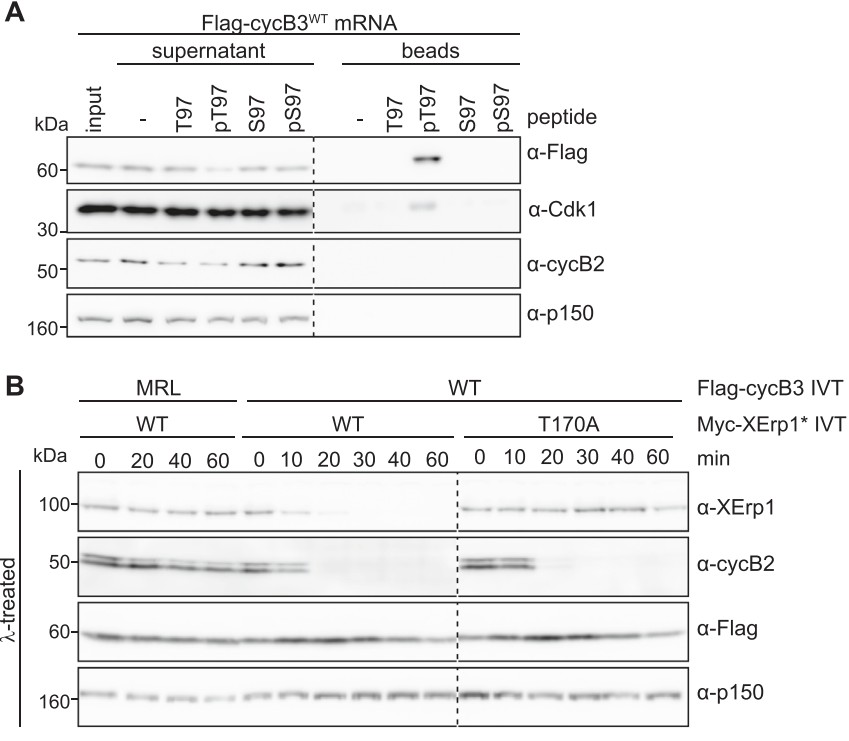

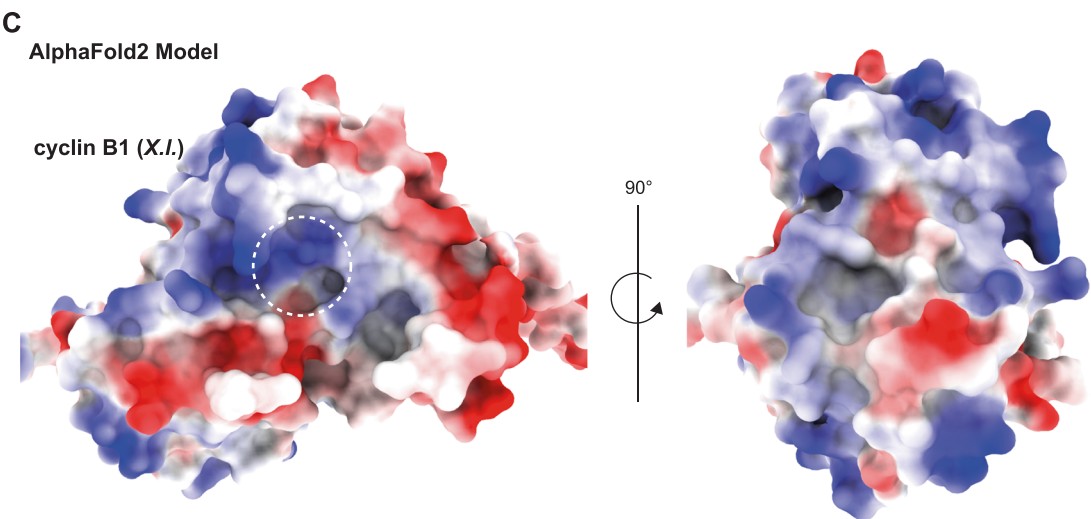

**Figure EV3. CycB3 phosphate-binding pocket interacts only with XErp1 pT97 but not pS97, XErp1 T170A mutation influence cyclin B3-induced XErp1 degradation and surface charge map of cyclin B1.**

(A) Bead-coupled XErp1 peptides (aa 90–104) containing unphosphorylated T97 (T97), phosphorylated T97 (pT97), unphosphorylated S97 (S97), or phosphorylated S97 (pS97) were incubated in CSF extract supplemented with mRNA encoding Flag-cyclin B3^WT. Exit from MII was prevented by the addition of MG262 and IVT XErp1^CT. Input, supernatant, and pull-downed beads were immunoblotted. p150 served as loading control. One representative experiment from three biological replicates is shown. (B) CSF extract was supplemented with IVT Myc-XErp1* WT or T170A variants and Flag-cyclin B3^WT/MRL. XErp1* is deficient in APC/C inhibition (see EV2C) to prevent that it interferes with meiotic exit. At indicated time points, samples were taken, treated with lambda phosphatase (λ) and immunoblotted. p150 served as loading control. (C) Surface charge map of AlphaFold2 model of *Xenopus* cyclin B1 in the corresponding position as cyclin B3 shown in Fig. 6B. White dashed circle highlights the phosphate binding pocket.

