## [Peer Review File · EMBO Reports]

A phosphate-binding pocket in cyclin B3 is essential for XErp1/Emi2 degradation in meiosis I

Rebecca Schunk, Marc Halder, Michael Schäfer, Elijah Johannes, Andreas Heim, Andreas Boland, and Thomas Mayer

Corresponding author(s): Thomas Mayer (thomas.u.mayer@uni-konstanz.de)

Review Timeline:

Submission Date:	3rd May 24
Editorial Decision:	1st Jul 24
Revision Received:	12th Sep 24
Editorial Decision:	31st Oct 24
Revision Received:	14th Nov 24
Accepted:	25th Nov 24

Editor: Deniz Senyilmaz Tiebe

Transaction Report:

Dear Thomas,

Thank you for submitting your research manuscript to our journal, which was now seen by three referees, whose reports are copied below.

I apologize for this unusual delay in getting back to you. It took longer than anticipated to receive the full set of referee reports.

Referees express interest in the proposed mechanism by which XErp1/Emi2 is degraded in meiosis I. However, they also raise some concerns that need to be addressed to consider publication here.

I find the reports informed and constructive, and believe that addressing the concerns raised will significantly strengthen the manuscript. As the reports are below, and I think all points need to be addressed, I will not detail them here.

Given these positive recommendations, we would like to invite you to submit a revised manuscript. Please revise your manuscript with the understanding that the referee concerns (as in their reports) must be fully addressed and their suggestions taken on board. Please address all referee concerns in a complete point-by-point response. Acceptance of the manuscript will depend on a positive outcome of a second round of review. It is EMBO reports policy to allow a single round of major experimental revision only and acceptance or rejection of the manuscript will therefore depend on the completeness of your responses included in the next, final version of the manuscript.

We realize that it is difficult to revise to a specific deadline. In the interest of protecting the conceptual advance provided by the work, we recommend a revision within 3 months. Please discuss the revision progress ahead of this time with me if you require more time to complete the revisions, or if you have questions or comments regarding the revision (also by video chat).

1. A data availability section providing access to data deposited in public databases is missing (where applicable).
2. Your manuscript contains statistics and error bars based on $n=2$. Please use scatter plots in these cases.

You can submit the revision either as a Scientific Report or as a Research Article. For Scientific Reports, the revised manuscript can contain up to 5 main figures and 5 Expanded View figures, and it should not exceed 27000 characters. If the revision leads to a manuscript with more than 5 main figures it will be published as a Research Article. In this case the Results and Discussion section should be separate. If a Scientific Report is submitted, these sections have to be combined. This will help to shorten the manuscript text by eliminating some redundancy that is inevitable when discussing the same experiments twice. In either case, all materials and methods should be included in the main manuscript file.

3) We replaced Supplementary Information with Expanded View (EV) Figures and Tables that are collapsible/expandable online. A maximum of 5 EV Figures can be typeset. EV Figures should be cited as 'Figure EV1, Figure EV2' etc... in the text and their respective legends should be included in the main text after the legends of regular figures.

4) a .docx formatted letter INCLUDING the reviewers' reports and your detailed point-by-point responses to their comments. As part of the EMBO publication's Transparent Editorial Process, EMBO reports publishes online a Review Process File (RPF) to

accompany accepted manuscripts. This File will be published in conjunction with your paper and will include the referee reports, your point-by-point response and all pertinent correspondence relating to the manuscript.

<https://www.embopress.org/page/journal/14693178/authorguide#transparentprocess>

5) a complete author checklist, which you can download from our author guidelines

<https://www.embopress.org/page/journal/14693178/authorguide>. Please insert information in the checklist that is also reflected in the manuscript. The completed author checklist will also be part of the RPF.

6) Please note that all corresponding authors are required to supply an ORCID ID for their name upon submission of a revised manuscript (<<https://orcid.org/>>). Please find instructions on how to link your ORCID ID to your account in our manuscript tracking system in our Author guidelines

<<https://www.embopress.org/page/journal/14693178/authorguide#authorshipguidelines>>

Additional information on source data and instruction on how to label the files are available:

<https://www.embopress.org/page/journal/14693178/authorguide#sourcedata>

9) Our journal encourages inclusion of *data citations in the reference list* to directly cite datasets that were re-used and obtained from public databases. Data citations in the article text are distinct from normal bibliographical citations and should directly link to the database records from which the data can be accessed. In the main text, data citations are formatted as follows: "Data ref: Smith et al, 2001" or "Data ref: NCBI Sequence Read Archive PRJNA342805, 2017". In the Reference list, data citations must be labeled with "[DATASET]". A data reference must provide the database name, accession number/identifiers and a resolvable link to the landing page from which the data can be accessed at the end of the reference. Further instructions are available at <http://www.embopress.org/page/journal/14693178/authorguide#referencesformat>

10) Regarding data quantification (see Figure Legends:

<https://www.embopress.org/page/journal/14693178/authorguide#figureformat>)

11) The journal requires a statement specifying whether or not authors have competing interests (defined as all potential or actual interests that could be perceived to influence the presentation or interpretation of an article). In case of competing

interests, this must be specified in your disclosure statement. Further information: <https://www.embopress.org/competing-interests>

12) Please also note our reference format:

13) All Materials and Methods need to be described in the main text using our 'Structured Methods' format, which is required for all research articles. According to this format, the Methods section includes a Reagents and Tools Table (listing key reagents, experimental models, software and relevant equipment and including their sources and relevant identifiers) followed by a Methods and Protocols section describing the methods using a step-by-step protocol format. The aim is to facilitate adoption of the methodologies across labs. More information on how to adhere to this format as well as a downloadable template (.docx) for the Reagents and Tools Table can be found in our author guidelines:

I look forward to seeing a revised version of your manuscript when it is ready. Please let me know if you have questions or comments regarding the revision.

Kind regards,

Deniz

Deniz Senyilmaz Tiebe, PhD
Scientific Editor
EMBO Reports

Referee #1:

XErp1/Emi2 inhibits the APC/C E3 ligase and mediates metaphase II arrest of oocytes before fertilization. It needs to be degraded in meiosis I. How XErp1/Emi2 is degraded in MI is not fully understood. In this manuscript, Demmig et al. show that Cdk1/cyclin B3 binds to XErp1/Emi2 and facilitates its phosphorylation by Plk1 and subsequent proteasomal degradation in meiosis I. Cyclin B3 binds XErp1/Emi2 using a conserved phosphate-binding pocket and promotes hyperphosphorylation of XErp1/Emi2. In addition, the authors show that this mechanism is evolutionarily conserved in *Xenopus* and mice. This study provides a detailed explanation of the timely and tight control of XErp1/Emi2 in meiosis I.

Overall, the data presented are solid and convincing. The study provides new insights into the regulation of XErp1/Emi2, a key factor of meiotic progression. It should be published in a major journal. The following questions need to be addressed prior to publication.

1) To test the binding between cyclin B3 and phosphorylated T97 from XErp1, the authors performed pull-down assays using a phosphopeptide from XErp1. What is the amino acid sequence of this XErp1 phosphopeptide? Since XErp1 can be phosphorylated at multiple sites, does the phosphate-binding pocket of cyclin B3 bind other phosphopeptides from XErp1, aside from the phosphorylated T97 peptide? If not, how does this phosphate-binding pocket achieve its specificity towards the pT97 peptide?

2) Phosphorylation of XErp1 T97 by Cdk1/cyclin B3 is required to target XErp1 for degradation and it is a prerequisite for the recruitment of Plk1. How does pT97 promote Plk1 recruitment? Does Plk1 bind directly to XErp1 pT97 or does it bind to another site? Is that site known?

3) How do XErp1 pT97 binding by cyclin B3 and phosphorylation by Plk1 promote its degradation? Do the multiple phosphorylation sites of XErp1 promote its binding and recognition by the E3 ligase?

4) The authors may wish to include a figure panel to show the domain and motifs of XErp1/Emi2 and its predicted AlphaFold structure with T97 and potential Plk1 phosphorylation sites highlighted. This will help to orient the readers about the locations of these phosphorylation sites.

Referee #2:

This study consists of *Xenopus* egg extracts and oocyte experiments demonstrating how CycB3/Cdk1 controls XErp1 (Emi2) degradation in M1. It turns out that the CycB3/Cdk1 XErp1 interaction is analogous to the Separase CycB1/Cdk1 interaction via a conserved phosphate binding pocket in CycB3. CycB3/Cdk1 phosphorylates XErp1 at Thr97, this residue is then a substrate for CycB3 binding and this interaction leads to multi-site XErp1 phosphorylation. This is required for Plk1 interaction and, ultimately, for the generation of phospho-degrons that lead to the SCF-dependent degradation of Emi1 in M1. This mechanism is critical to prevent a premature CSF arrest in M1 oocytes and allow progression towards M2 where CycB3 disappears and XErp1 can now accumulate to inhibit the APC/C and arrest the cell cycle until the incoming sperm triggers exit from M2.

This is a critical mechanistic feature of meiotic cell cycle control. Therefore, the paper makes an important contribution to the field and should be published.

The quality of the experiments is excellent. The XErp1 degradation assays in CSF extracts supplied with CycB3 mRNA are a great way to analyse the features of the CycB3 XErp1 interaction, and the experiments in Figs. 1, 2, and 3 make a convincing argument that supports the proposed model. A particularly strong experiment is the fusion of CycB3 and XErp1, which bypasses the phosphate binding pocket requirement (Fig. 3D).

Figure 4 demonstrates that the same mechanism applies to mouse CycB3, suggesting evolutionary conservation.

Figure 5 then tests the hypothesis under physiological conditions using *Xenopus* oocytes and an element degron approach to deplete endogenous CycB3. This allows a rigorous test of the importance of the phosphate binding pocket in CycB3 for M1 exit.

Experimentally, I do not think the authors need to add anything to this study. The paper makes an exceptionally strong case to support the proposed model.

I only have a few comments on the text that may help to improve the readability of the MS and highlight some important aspects of the study.

1) The introduction doesn't follow a logical narrative. For example, the switch from paragraph 1 (an introduction of CycB3 and meiosis) to paragraph 2 (A discussion of the CycB1 Separate interaction) is too abrupt and confusing for the reader.

2) The experiment in Figure 1D shows that CycB3 interacts with phosphorylated Threonines. The major band that is shown here is ~ 160 kDa, and the blots shown are only very high molecular weight proteins. Do the authors see a band at 90kDa that could correspond to XErp1? The authors should show the entire blot! The high molecular weight band should be discussed in the result description. There is a brief mention of this at the end of the discussion, but this is an important set of data that points to additional interactions of CycB3. This should be highlighted.

3) Figure 5D: The pattern of XErp1 accumulation is not completely clear here. Why is XErp1 degraded after M1 in the CycB3 depletion condition, yet it accumulates in M2 when cycB3PMD15 is reconstituted? This should be discussed properly? Is CycB3 important for stabilisation of XErp1 in M2 as well as its degradation in M1?

Referee #3:

This study explores the regulation of the meiotic cell cycle in vertebrate oocytes, focusing on the mechanisms by which oocytes distinguish between meiosis I and meiosis II to arrest at metaphase II for fertilization. This arrest is crucial to prevent aneuploid embryos, which are often linked to infertility and recurrent miscarriages in humans. Understanding the molecular basis for the arrest at metaphase II rather than metaphase I is therefore an important question in sexual reproduction.

Previous research by the team has shown that this process depends on tight regulation of Cdk1-Cyclin B3. In meiosis I, Cdk1-Cyclin B3 phosphorylates an APC inhibitor, XErp1/Emi2, at T97, thereby inducing its degradation in metaphase I. Without Cyclin B3, XErp1 accumulates prematurely at metaphase I, preventing APC activation and causing early arrest, which compromises fertilization. Despite knowing the consequences of Cyclin B3 silencing on XErp1 stability and meiotic progression, the detailed mechanism of XErp1 degradation remains unclear.

Using *in vitro* approaches in *Xenopus* metaphase II-arrested extracts (CSF extracts) and *in vivo* experiments in *Xenopus* oocytes, the authors discovered that XErp1 degradation depends not only on phosphorylation at T97 by Cdk1-Cyclin B3 but also on a novel determinant of Cyclin B3: the phosphate-binding pocket (PBP). The PBP enables Cdk1-Cyclin B3 to interact with substrates that are already phosphorylated on Threonine residues but distinct from Cyclin B1, including XErp1. This interaction favours the phosphorylation of XErp1 at "low CDK affinity sites", which are then required for its degradation. Accordingly, in the absence of the PBP, Cdk1-Cyclin B3 can no longer induce XErp1 degradation in CSF extracts. Moreover, Cyclin B3-depleted *Xenopus* oocytes are unable to exit from the metaphase I arrest upon expression of a PBP mutant of Cyclin B3, in contrast to wild-type Cyclin B3.

From these results, the authors propose an attractive stepwise mechanism for XErp1 degradation controlled by Cdk1-Cyclin B3. Cdk1-Cyclin B3 initially phosphorylates XErp1 at T97, independently of the PBP. The PBP then binds to phosphorylated T97, thereby recruiting Cyclin B3 to XErp1. As a result, this process allows further phosphorylation of XErp1 by Cdk1-Cyclin B3, including a potential site recognized by Plk1 although this remains to be characterised. Subsequently, Plk1 is recruited to XErp1 and induces its degradation. Importantly, this mechanism is specific to Cyclin B3, indicating that the PBP within B-type Cyclins

contributes to their functional specificities.

To sum-up, the manuscript reports a single key finding, which is the stepwise mechanism of XErp1 controlled by Cdk1-Cyclin B3, a process preventing premature arrest in metaphase I of vertebrate oocytes. These findings provide deeper insights into the regulation of female meiosis and open new avenues for understanding how Cyclin B1 and Cyclin B3 have distinct functions during meiotic cell cycle. These findings will be of interest to the scientific research community working in the cell cycle field. The experiments are well designed and the conclusions are convincingly supported by the results. However, I have some recommendations to improve the quality of the manuscript and additional experiments needed for publication.

Major comments

General considerations

However, due to the complex mechanism studied, a schematic in the main figures illustrating the proposed model is recommended. Reorganizing the paper to follow the proposed model would also avoid backtracking and enhance readability of the manuscript:

1. Characterization of the PBP
2. Function of the PBP on XErp1 degradation in vitro and in vivo
3. Mechanism dissection following the proposed model:
 - a. Phosphorylation of T97
 - b. Additional phosphorylation of XErp1 mediated by the PBP of Cyclin B3
 - c. Binding to Plk1 (currently in supplementary data; should be in main figures as Plk1 is the kinase that ultimately drives XErp1 degradation through the two degron motifs)
4. Evolutionary conservation of the mechanism using mouse proteins

Major comments on Results

1. Figs 1C and 1D: the rationale for using two different conditions (APC active and APC inhibited) is unclear. For both conditions, a comparative analysis of Separase and endogenous XErp1 association to either Cyclin B1 or Cyclin B3 (WT or mutant) should be provided. For Fig. 1D (APC inhibited), additional western blots using pSer antibodies must be included to support the interaction specificity of Cyclin B3 with phosphorylated proteins, in particular with pThr proteins. If only pThr proteins associated, the authors could argue about specificity of Cyclins (Threonine versus Serine). Many commercial antibodies are available to probe the supernatants and the beads. Since some of them recognize phospho-sites targeted by both CDK and MAPK activities, the reaction mix can be supplemented with a MAPK inhibitor (U0126), similar to the use of λ phosphatase in Fig. 1D.
2. Fig. 2D: This experiment is difficult to understand because the kinase used to phosphorylate XErp1 is not specified. Is it Cdk1-Cyclin B1 or Cdk1-Cyclin B3? In addition, it is unclear why XErp1 T97A is phosphorylated weakly despite T97 being crucial for PBP association. XErp1 T97A is phosphorylated, albeit weakly, whereas this site is required for the association of the PBP with XErp1. This section needs clearer explanation.
3. Fig. 2E (and E2VD): The authors concluded that "the phosphorylation of XErp1 at T97 by Cdk1-Cyclin B3 is required to target XErp1 for degradation as it is a prerequisite for the recruitment of Plk1. However, it is not sufficient to destabilize XErp1, as phosphorylations at additional Cdk1 (S/T)P sites are necessary". To strengthen their conclusions, additional results should be included showing that pulled-down XErp1 (WT or mutants) from CSF extracts (blocked in Metaphase II) expressing either Cyclin B3WT or Cyclin B3PM, associates with Plk1 and is phosphorylated at multiple sites by western blots using pThr or pSer antibodies, and, if available, pT170 antibodies. It would also be of great interest to monitor the stability and phosphorylation state of different forms of XErp1 (e.g. T170A mutant and in vitro T97-thiophosphorylated XErp1WT or XErp17A).
4. Fig. 3B: "PBP binds to phosphorylated T97". It would be valuable to demonstrate that pT97 peptides compete with the mechanism controlled by Cdk1-Cyclin B3 in promoting the phosphorylation and degradation of endogenous XErp1. In other words, can these peptides prevent the exit of CSF extracts following the expression of Cyclin B3WT? This would strengthen the conclusion that the PBP of Cyclin B3 behaves as a "docking site" for pre-phosphorylated proteins including XErp1.
5. Fig. 5B: there is no clear indication of the conditions illustrated (Cyclin B3Ab, Cyclin B3Ab + Cyclin B3WT or Cyclin B3Ab + Cyclin B3PM).
6. Fig. 5D: This in vivo experiment is crucial to the paper as it addresses the function of the PBP of Cyclin B3 in *Xenopus* oocytes. It demonstrates that the PBP of Cyclin B3 is essential to prevent the accumulation of XErp1 in meiosis I, thereby preventing a premature arrest in metaphase I. However, the western blots of endogenous XErp1 and Cyclin B3 are very faint. In Cyclin B3-depleted oocytes (CycB3Ab), XErp1 appears to be degraded 4 hours after GVBD, suggesting that exit from metaphase I is not abolished but only delayed. Moreover, the disappearance of ectopically expressed Cyclin B3 (WT Δ 15 or PM Δ 15) signals 4 hours after GVBD need explanation as it suggests that Cyclin B3 is degraded (which takes place during meiosis II). An additional marker beyond APC3 phosphorylation is required to confirm meiotic progression. A Cyclin B2 western blot would confirm that Cyclin B3-depleted oocytes, injected or not with CycB3PM Δ 15, remain arrested in metaphase I as Cyclin B2 is not degraded after GVBD. A kinase assay would also be useful. It is puzzling that although oocytes were injected with both Flag-TRIM21 and Flag-Cyclin B3, the Flag western blot shows only one signal.
7. Fig. EV1B and EV1C: Based on what is known about Cdk1-Cyclin B1-Cks1, the authors depleted Cks2 and found that Cyclin B3 triggers exit from metaphase II arrest independently of Cks2. It is unclear why the authors focused on Cks2 without analysing Cks1. In a paper from 2014 (10.1016/j.cub.2014.05.044), Cks1 was identified in metaphase II-arrested oocytes using proteomic, but at much lower concentrations compared to Cks2. This must be discussed by the authors.

8. Material and methods:

- Include descriptions for antibodies against pThr (with the sequence of the phospho-site) and Separase.
- Describe the phosphorylation protocol for XErp1NT and T97 peptides with the kinase that have been used.

Minor comments

The manuscript should be edited for consistency and clarity. Unnecessary details, such as protein tags, should be omitted.

Please, find below some examples for the Figures:

- Use consistent terminology (e.g., Cdc27/APC3).
- Correct labeling and include relevant details (e.g., "GVB" to "GVBD" in Fig. 5C, dashes missing between "Pro I" and "1h" in timelines of Fig. 5D).
- Maintain consistency in labeling across main and supplementary figures (e.g., Flag-Cyclin B3WT or Flag-Cyclin B3PM in all figures).

First, we want to thank all three referees for their positive and enthusiastic evaluation of our manuscript. Their insightful comments and expert suggestions greatly improved this study. As detailed in our point-by-point response, we have addressed nearly all the issues raised by the reviewers. The new data included in our revised manuscript significantly strengthen our findings. Below is our point-by-point response to their specific comments.

Just a quick note: The author list has been updated from Rebecca Demmig to Rebecca Schunk, as the first author got married and adopted her husband's last name.

Referee #1:

XErp1/Emi2 inhibits the APC/C E3 ligase and mediates metaphase II arrest of oocytes before fertilization. It needs to be degraded in meiosis I. How XErp1/Emi2 is degraded in MI is not fully understood. In this manuscript, Demmig et al. show that Cdk1/cyclin B3 binds to XErp1/Emi2 and facilitates its phosphorylation by Plk1 and subsequent proteasomal degradation in meiosis I. Cyclin B3 binds XErp1/Emi2 using a conserved phosphate-binding pocket and promotes hyperphosphorylation of XErp1/Emi2. In addition, the authors show that this mechanism is evolutionarily conserved in *Xenopus* and mice. This study provides a detailed explanation of the timely and tight control of XErp1/Emi2 in meiosis I.

Overall, the data presented are solid and convincing. The study provides new insights into the regulation of XErp1/Emi2, a key factor of meiotic progression. It should be published in a major journal. The following questions need to be addressed prior to publication.

We thank the reviewer for his positive feedback and constructive comments. We have addressed the comments as detailed below, and feel that the changes and experimental additions have greatly improved the manuscript.

1) To test the binding between cyclin B3 and phosphorylated T97 from XErp1, the authors performed pull-down assays using a phosphopeptide from XErp1. What is the amino acid sequence of this XErp1 phosphopeptide?

This is an important point. We added the peptide sequence in the method section of the revised version of our manuscript.

Since XErp1 can be phosphorylated at multiple sites, does the phosphate-binding pocket of cyclin B3 bind other phosphopeptides from XErp1, aside from the phosphorylated T97 peptide? If not, how does this phosphate-binding pocket achieve its specificity towards the pT97 peptide?

We thank the reviewer for highlighting this point. In the original version of the manuscript, we speculated that the negatively charged surface in the proximity of cyclin B3's phosphate-binding pocket contributes to specificity by interacting with three basic arginine residues just C-terminal to XErp1 pT97 (Fig. A).

T97: **TP**RVGKKEFSL**RRR**LLISKA
 S43: **SP**D SHKSGNFLETVTEGYEN
 S73: **SP**IKYELSWGADTRESKQLA
 S157: **SP**RDGSYEP IATSTLKTESE
 S213: **SP**VQHSLASSTDDSI LYEET
 T239: **TP**TCNFIVKEEFQTPISNLA
 T252: **TP**ISNLAANFRFNLCTPDVG
 T267: **TP**DVGHVSDFDISVTEDSA F

Fig. A: Amino acid sequences C-terminal to the different (S/T)P sites in XErp1 critical for XErp1 degradation. The different (S/T)P sites and the three arginine residues C-terminal to T97 are shown in bold.

For the revised version of our manuscript, we addressed this hypothesis experimentally. Specifically, we analyzed if mutation of the three arginine residues to alanines located C-terminal of T97 affects the association of XErp1 with cyclin B3. Indeed, co-immunoprecipitation (coIP) experiments revealed that XErp1 with three arginine to alanine mutations (3RA) – like the non-phosphorylatable T97A mutant – failed to co-precipitate with cyclin B3^{WT}. These novel data are shown in the **new** figure 6C.

These data suggest that the positively charged arginine residues in proximity to XErp1 T97 are critical for the interaction with cyclin B3. If this applies, the 3RA mutant should be stabilized compared to WT XErp1 in the presence of cyclin B3^{WT}. Indeed, as shown in the **new** figure 6D, the cyclin B3-mediated degradation of XErp1 is significantly slowed down when XErp1 carries the 3RA mutations. Of note, the other (S/T)P sites critical for XErp1 degradation are not followed by a stretch of positively charged amino acids (see Fig. A).

In sum, these new data strongly support our hypothesis that the negatively charged surface surrounding cyclin B3's phosphate-binding pocket contributes to specificity by interacting with the positively charged arginine residues in XErp1. As discussed in the manuscript, the phosphate-binding pocket of cyclin B1 is located within a positively charged surface area explaining differential substrate specificity of cyclin B3's and cyclin B1's phosphate-binding pocket.

2) Phosphorylation of XErp1 T97 by Cdk1/cyclin B3 is required to target XErp1 for degradation and it is a prerequisite for the recruitment of Plk1. How does pT97 promote Plk1 recruitment? Does Plk1 bind directly to XErp1 pT97 or does it bind to another site? Is that site known?

We appreciate the reviewer for bringing this up. As shown by Michel Yaffe's team¹ and confirmed by subsequent publications from many different labs, the polo-box domain (PBD) of Plk1 has an "extremely high" preference for a serine residue at position -1 preceding the phosphorylated serine or threonine residue. As shown below, XErp1 T97 is not preceded by a serine residue making it unlikely that Plk1 binds directly to T97-phosphorylated XErp1 (Fig. B).

T⁹⁷P

X. laevis	...ETPR...
X. tropicalis	...ETPR...
H. sapiens	...ETPK...
M. musculus	...ETPK...
R. norvegicus	...ETPK...
D. rerio	...ETPK...
F. catus	...ETPK...
C. lupus	...ETPK...
O. latipes	...ETPR...
M. mulatta	...ETPK...

Fig. B: Sequence alignment of XErp1/Emi2 from different species showing that T97 is not preceded by a serine residue.

XErp1 contains two known binding sites for Plk1's polo box domain, i.e., ST¹⁷⁰ and ST¹⁹⁵. As shown by us previously², only ST¹⁷⁰ is essential for the cyclin B3-mediated degradation of XErp1. To demonstrate that ST¹⁷⁰ is critical for Plk1 binding, we performed co-immunoprecipitation experiments. Indeed, as shown in the **new** figure 5C, Plk1 binding to XErp1 in the presence of cyclin B3^{WT} is lost upon mutation of T¹⁷⁰ to alanine. These novel data nicely support our data regarding the role of T¹⁷⁰ for Plk1 binding.

3) How do XErp1 pT97 binding by cyclin B3 and phosphorylation by Plk1 promote its degradation? Do the multiple phosphorylation sites of XErp1 promote its binding and recognition by the E3 ligase?

For XErp1 three degradative mechanisms are known: First, its efficient degradation upon fertilization involving CaMKII and Plk1^{3,4}. Second, its slow degradation during the metaphase II arrest of mature eggs involving Cdk1/cyclin B1 and Plk1^{5,6}. Third, its efficient degradation in MI involving Cdk1/cyclin B3 and Plk1². In all these cases, Plk1 upon its recruitment to XErp1 phosphorylates the two phospho-degrons (DSG₃S³⁸ and DSA₂S²⁸⁸), which are then recognized by the E3 ligase SCF^{βTRCP} resulting in the proteasomal degradation of XErp1. We explain this mechanism in more detail in the revised version of our manuscript.

4) The authors may wish to include a figure panel to show the domain and motifs of XErp1/Emi2 and its predicted Alphafold structure with T97 and potential Plk1 phosphorylation sites highlighted. This will help to orient the readers about the locations of these phosphorylation sites.

We thank the reviewer for this helpful comment and changed Fig. EV2C accordingly. Due to the unstructured nature of XErp1 we did not include AlphaFold structures of the different XErp1 domains.

Referee #2:

This study consists of *Xenopus* egg extracts and oocyte experiments demonstrating how CycB3/Cdk1 controls Xerp1 (Emi2) degradation in M1. It turns out that the CycB3/Cdk1 Xerp1 interaction is analogous to the Separase CycB1/Cdk1 interaction via a conserved phosphate binding pocket in CycB3. CycB3/Cdk1 phosphorylates Xerp1 at Thr97, this residue is then a substrate for CycB3 binding and this interaction leads to multi-site Xerp1 phosphorylation. This is required for Plk1 interaction and, ultimately, for the generation of phospho-degrons that lead to the SCF-dependent degradation of Emi1 in M1. This mechanism is critical to prevent a premature CSF arrest in M1 oocytes and allow progression towards M2 where CycB3 disappears and Xerp1 can now accumulate to inhibit the APC/C and arrest the cell cycle until the incoming sperm triggers exit from M2.

This is a critical mechanistic feature of meiotic cell cycle control. Therefore, the paper makes an important contribution to the field and should be published.

The quality of the experiments is excellent. The Xerp1 degradation assays in CSF extracts supplied with CycB3 mRNA are a great way to analyse the features of the CycB3 Xerp1 interaction, and the experiments in Figs. 1, 2, and 3 make a convincing argument that supports the proposed model. A particularly strong experiment is the fusion of CycB3 and Xerp1, which bypasses the phosphate binding pocket requirement (Fig. 3D). Figure 4 demonstrates that the same mechanism applies to mouse CycB3, suggesting evolutionary conservation.

Figure 5 then tests the hypothesis under physiological conditions using *Xenopus* oocytes and an element degron approach to deplete endogenous CycB3. This allows a rigorous test of the importance of the phosphate binding pocket in CycB3 for M1 exit.

Experimentally, I do not think the authors need to add anything to this study. The paper makes an exceptionally strong case to support the proposed model. I only have a few comments on the text that may help to improve the readability of the MS and highlight some important aspects of the study.

We are delighted by the reviewer's enthusiastic feedback and are thankful for her/his comments to improve the quality and readability of the manuscript. The changes and experimental additions have greatly improved the manuscript.

1) The introduction doesn't follow a logical narrative. For example, the switch from paragraph 1 (an introduction of CycB3 and meiosis) to paragraph 2 (A discussion of the CycB1 Separate interaction) is too abrupt and confusing for the reader.

We thank the reviewer for highlighting this point. We changed the text accordingly to make the transition from the first to the second paragraph easier to read.

2) The experiment in Figure 1D shows that CycB3 interacts with phosphorylated Threonines. The major band that is shown here is $>160\text{kDa}$, and the blots shown are only very high molecular weight proteins. Do the authors see a band at 90kDa that could correspond to XErp1? The authors should show the entire blot! The high molecular weight band should be discussed in the result description. There is a brief mention of this at the end of the discussion, but this is an important set of data that points to additional interactions of CycB3. This should be highlighted.

We thank the reviewer for this comment. In the revised version, we show the entire blot for pThr (Fig. 1D). However, we do not detect a specific band that could be assigned to XErp1. Yet, this might not be surprising given that the concentration of XErp1 in CSF extract is roughly 14nM, compared to 4,700nM for α -4A tubulin⁷. As suggested, we discuss the high molecular weight bands in the result section of our revised manuscript.

3) Figure 5D: The pattern of XErp1 accumulation is not completely clear here. Why is Xerp1 degraded after M1 in the CycB3 depletion condition, yet it accumulates in M2 when cycB3^{PM Δ 15} is reconstituted? This should be discussed properly? Is CycB3 important for stabilisation of XErp1 in M2 as well as its degradation in M1?

We thank the reviewer for this comment, which was also raised by reviewer #3. To rigorously address this point, we repeated the oocyte depletion/rescue experiment three more times. As shown in the *new* figure 2E, protein levels of XErp1 and cyclin B3 now show the expected behavior. Specifically, XErp1 accumulates prematurely (GVBD+1h), and its levels continue to increase (GVBD+4h) in cyclin B3 depleted oocytes not expressing a rescue construct or expressing pocket mutant cyclin B3 (cycB3^{PM Δ 15}). In control depleted oocytes or cyclin B3 depleted oocytes expressing the WT rescue construct (cycB3^{WT Δ 15}), XErp1 does not efficiently accumulate until GVBD+4h. Since these results were highly reproducible, we decided to exchange the old figure with the new one.

Referee #3:

This study explores the regulation of the meiotic cell cycle in vertebrate oocytes, focusing on the mechanisms by which oocytes distinguish between meiosis I and meiosis II to arrest at metaphase II for fertilization. This arrest is crucial to prevent aneuploid embryos, which are often linked to infertility and recurrent miscarriages in humans. Understanding the molecular basis for the arrest at metaphase II rather than metaphase I is therefore an important question in sexual reproduction.

Previous research by the team has shown that this process depends on tight regulation of Cdk1-Cyclin B3. In meiosis I, Cdk1-Cyclin B3 phosphorylates an APC inhibitor, XErp1/Emi2, at T97, thereby inducing its degradation in metaphase I. Without Cyclin B3, XErp1 accumulates prematurely at metaphase I, preventing APC activation and causing

early arrest, which compromises fertilization. Despite knowing the consequences of Cyclin B3 silencing on XErp1 stability and meiotic progression, the detailed mechanism of XErp1 degradation remains unclear.

Using in vitro approaches in *Xenopus* metaphase II-arrested extracts (CSF extracts) and in vivo experiments in *Xenopus* oocytes, the authors discovered that XErp1 degradation depends not only on phosphorylation at T97 by Cdk1-Cyclin B3 but also on a novel determinant of Cyclin B3: the phosphate-binding pocket (PBP). The PBP enables Cdk1-Cyclin B3 to interact with substrates that are already phosphorylated on Threonine residues but distinct from Cyclin B1, including XErp1. This interaction favours the phosphorylation of XErp1 at "low CDK affinity sites", which are then required for its degradation. Accordingly, in the absence of the PBP, Cdk1-Cyclin B3 can no longer induce XErp1 degradation in CSF extracts. Moreover, Cyclin B3-depleted *Xenopus* oocytes are unable to exit from the metaphase I arrest upon expression of a PBP mutant of Cyclin B3, in contrast to wild-type Cyclin B3.

From these results, the authors propose an attractive stepwise mechanism for XErp1 degradation controlled by Cdk1-Cyclin B3. Cdk1-Cyclin B3 initially phosphorylates XErp1 at T97, independently of the PBP. The PBP then binds to phosphorylated T97, thereby recruiting Cyclin B3 to XErp1. As a result, this process allows further phosphorylation of XErp1 by Cdk1-Cyclin B3, including a potential site recognized by Plk1 although this remains to be characterised. Subsequently, Plk1 is recruited to XErp1 and induces its degradation. Importantly, this mechanism is specific to Cyclin B3, indicating that the PBP within B-type Cyclins contributes to their functional specificities. To sum-up, the manuscript reports a single key finding, which is the stepwise mechanism of XErp1 controlled by Cdk1-Cyclin B3, a process preventing premature arrest in metaphase I of vertebrate oocytes. These findings provide deeper insights into the regulation of female meiosis and open new avenues for understanding how Cyclin B1 and Cyclin B3 have distinct functions during meiotic cell cycle. These findings will be of interest to the scientific research community working in the cell cycle field. The experiments are well designed and the conclusions are convincingly supported by the results. However, I have some recommendations to improve the quality of the manuscript and additional experiments needed for publication.

We thank the reviewer for her/his positive feedback, the thorough analyses of our study and the resulting constructive comments. We have addressed the points raised by the reviewer (see below) and the additional experiments and text changes significantly improved our manuscript.

Major comments

General considerations

However, due to the complex mechanism studied, a schematic in the main figures illustrating the proposed model is recommended. Reorganizing the paper to follow the

proposed model would also avoid backtracking and enhance readability of the manuscript:

1. Characterization of the PBP
2. Function of the PBP on XErp1 degradation in vitro and in vivo
3. Mechanism dissection following the proposed model:
 - a. Phosphorylation of T97
 - b. Additional phosphorylation of XErp1 mediated by the PBP of Cyclin B3
 - c. Binding to Plk1 (currently in supplementary data; should be in main figures as Plk1 is the kinase that ultimately drives XErp1 degradation through the two degron motifs)
4. Evolutionary conservation of the mechanism using mouse proteins

We thank the reviewer for pointing this out. To adequately address these points, we completely re-structured the whole manuscript following the outline suggested by the reviewer. Since these changes affected the whole manuscript without adding novel data, we decided against highlighting them using the track changes modus (otherwise the manuscript would be difficult to read). The new structure makes the manuscript much easier to read and we are, therefore, thankful for the reviewer's comment.

Major comments on Results

1. Figs 1C and 1D: the rationale for using two different conditions (APC active and APC inhibited) is unclear. For both conditions, a comparative analysis of Separase and endogenous XErp1 association to either Cyclin B1 or Cyclin B3 (WT or mutant) should be provided.

We agree with the reviewer that it is confusing to use different conditions for Figs. 1C and 1D. In our hands, endogenous separase could only be co-precipitated with cyclin B1 when the APC/C was active (MG262 + OA), but not when the APC/C was inactive (MG262 + XErp1^{CT}). While this observation was highly reproducible, we can only speculate about the underlying reason. We assume that endogenous securin and cyclin B get ubiquitylated in APC/C active CSF extract and that the ubiquitylation of endogenous securin and cyclin B, which does not result in their degradation due to the presence of MG262, negatively affects their binding to separase. A similar mechanism has been shown for CDC20, whose non-proteolytic ubiquitylation affects its association with spindle assembly checkpoint proteins⁸. In our experiment, reduced binding of (ubiquitylated) securin and cyclin B1 to separase would allow ectopic cyclin B1, which was added after pre-incubation of the extract with MG262 and OA, to compete more efficiently for separase binding.

Therefore, to harmonize the conditions used for the experiments shown in Figs. 1C and 1D, we used the condition with active APC/C (MG262 + OA). Thus, the *new* Fig. 1D, which shows one representative result of three independent experiments, is now directly comparable to Fig. 1C and confirms that cyclin B3 – although not being able to engage phosphorylated separase – possesses a functional phosphate-binding pocket. Since we agree with the reviewer that Fig. 1 should focus on the general characterization of cyclin

B3's PBP (see above), we have refrained to include data regarding XErp1 already at this point of the manuscript.

For Fig. 1D (APC inhibited), additional western blots using pSer antibodies must be included to support the interaction specificity of Cyclin B3 with phosphorylated proteins, in particular with pThr proteins. If only pThr proteins associated, the authors could argue about specificity of Cyclins (Threonine versus Serine). Many commercial antibodies are available to probe the supernatants and the beads. Since some of them recognize phospho-sites targeted by both CDK and MAPK activities, the reaction mix can be supplemented with a MAPK inhibitor (U0126), similar to the use of λ phosphatase in Fig. 1D.

We thank the reviewer for pointing this out. Following the rationale explained above, we performed the cyclin B3 IP from CSF extract treated with MG262 and okadaic acid (APC/C active) and analyzed the samples using three different commercially available pSer antibodies. In our hands, the antibody from Santa Cruz (sc-81514) gave the most reliable results. These novel data are shown in the **new** Fig. EV1B and suggest that cyclin B3 is able to bind proteins that are phosphorylated at serine residues. However, due to the presence of the phosphatase inhibitor OA we could not analyze if inhibition of MAPK affects the phosphorylation state of proteins associated with cyclin B3.

While these novel data suggest that cyclin B3 can bind proteins that are phosphorylated at serine residues, it does not provide any information if the phosphorylated serine residues are indeed responsible for the interaction with cyclin B3. To address this, we performed novel phosphopeptide pulldown assays using a XErp1 phosphopeptide where we replaced phosphorylated T97 (pT97) with a phosphorylated serine residue (pS97). These novel data show that cyclin B3 does not associate with the XErp1 pS97 phosphopeptide (**new** Fig. EV3A). Thus, these data suggest that cyclin B3's phosphate-binding pocket – at least in the context of XErp1 – preferentially binds phosphorylated threonine. We discuss this novel aspect in the revised version of our manuscript.

2. Fig. 2D: This experiment is difficult to understand because the kinase used to phosphorylate XErp1 is not specified. Is it Cdk1-Cyclin B1 or Cdk1-Cyclin B3? In addition, it is unclear why XErp1T97A is phosphorylated weakly despite T97 being crucial for PBP association. XErp1T97A is phosphorylated, albeit weakly, whereas this site is required for the association of the PBP with XErp1. This section needs clearer explanation.

We apologize for the confusion. As stated in the main text, the kinase used for the assay shown in Fig. 2D was recombinant Cdk1 in complex with either cyclin B3^{WT} or cyclin B3^{PM}. To make this clearer, we added this information to the figure legend.

As correctly stated by reviewer, XErp1 T97 is crucial for the association of cyclin B3 and, thus, for subsequent PBP-dependent phosphorylation events. Yet, there are also PBP-independent phosphorylation events as evident by the fact that XErp1^{NT} T97A, which is wildtype for the remaining seven Cdk1 sites, is radiolabeled in our in vitro kinase assays

(Fig. 5A). Thus, XErp1 is phosphorylated by Cdk1/cyclin B3 in a PBP-dependent and PBD-independent manner. We changed the text accordingly to explain this aspect in more detail.

3. Fig. 2E (and E2VD): The authors concluded that "the phosphorylation of XErp1 at T97 by Cdk1-Cyclin B3 is required to target XErp1 for degradation as it is a prerequisite for the recruitment of Plk1. However, it is not sufficient to destabilize XErp1, as phosphorylations at additional Cdk1 (S/T)P sites are necessary". To strengthen their conclusions, additional results should be included showing that pulled-down XErp1 (WT or mutants) from CSF extracts (blocked in Metaphase II) expressing either Cyclin B3^{WT} or Cyclin B3^{PM}, associates with Plk1 and is phosphorylated at multiple sites by western blots using pThr or pSer antibodies, and, if available, pT170 antibodies. It would also be of great interest to monitor the stability and phosphorylation state of different forms of XErp1 (e.g. T170A mutant and in vitro T97-thiophosphorylated XErp1^{WT} or XErp1^{7A}).

We thank the reviewer for pointing this out. To address this point, we performed XErp1 IPs from CSF extract supplemented with different XErp1 variants and WT or pocket mutant cyclin B3 and analyzed for the association of Plk1. As shown in the *new* Fig. 5C, Plk1 associates with XErp1 only under conditions where T97 and T170 are wildtype when incubated with wildtype, but not pocket mutant, cyclin B3. Unfortunately, both the pThr and pSer antibodies produced high background signals under these experimental conditions, preventing any meaningful conclusions. Furthermore, as suggested by the reviewer we created a pT170 antibody during the course of this project, which, regrettably, did not specifically detect XErp1 pT170. However, we believe that the phosphorylation shift of XErp1 can be used as proxy to estimate its phosphorylation. Consistent with the fact that binding of Plk1 to XErp1 requires wildtype T97, T170, as well as WT cyclin B3, XErp1 displays full phosphorylation only under these conditions (*new* Fig. 5C). We discuss this point in the revised version of our manuscript in more detail. Unfortunately, Cdk1/cyclin B3 belongs to the few kinases, which in our hand are inefficient in phosphorylating substrates in the presence of thio-ATP. We tried various conditions, but none of them gave the expected result. Parallel assays with a different kinase confirmed that this was not due to assay conditions. Thus, we were unable to test the stability of thio-phosphorylated XErp1 in CSF extract. However, to analyze the role of T170 for XErp1 degradation, we analyzed the stability of XErp1 T170A in CSF extract supplemented with cyclin B3^{WT}. These novel data, shown in the *new* Fig. EV3B, confirm that T170 is critical for the efficient degradation of XErp1 by Cdk1/cyclin B3.

4. Fig. 3B: "PBP binds to phosphorylated T97". It would be valuable to demonstrate that pT97 peptides compete with the mechanism controlled by Cdk1-Cyclin B3 in promoting the phosphorylation and degradation of endogenous XErp1. In other words, can these peptides prevent the exit of CSF extracts following the expression of Cyclin B3^{WT}? This would strengthen the conclusion that the PBP of Cyclin B3 behaves as a "docking site" for pre-phosphorylated proteins including XErp1.

We thank the reviewer for this great idea. We performed the suggested experiment and added the T97 or pT97 peptides to CSF extract supplemented with IVT cyclin B3^{WT}. Unfortunately, addition of the T97 or pT97 peptide unspecifically affected the quality of the CSF extract. We assume that the TFA (trifluoroacetate) salt, as which the peptides were delivered, accounted for this artificial effect.

Thus, to address the reviewer's comment by an alternative experiment we made use of a XErp1 fragment encompassing residues 91 to 126 (XErp1⁹¹⁻¹²⁶). Importantly, T97 is the only Cdk1 phosphorylation site within this XErp1 fragment. As shown in the **new** figure 4D, addition of WT XErp1⁹¹⁻¹²⁶, but not of T97A XErp1⁹¹⁻¹²⁶, significantly delayed the degradation of endogenous XErp1 in CSF extract supplemented with IVT cyclin B3^{WT}. Thus, as suggested by the reviewer these new data nicely support our model that the PBP of cyclin B3 serves as a "docking site" for pre-phosphorylated XErp1.

5. Fig. 5B: there is no clear indication of the conditions illustrated (Cyclin B3Ab, Cyclin B3Ab + Cyclin B3WT or Cyclin B3Ab + Cyclin B3PM).

We apologize for the confusion. The fluorescent images shown in Fig. 5B (now 2B) do not belong to one specific conditions, e.g., cyclin B3-depleted oocytes, but rather display representative examples of spindle and/or polar body structures we observed across all conditions. Based on these structures, we categorized the cell cycle state of oocytes in MI and post-MI. The quantification is shown in Fig. 2C. We revised the text to clarify this point.

6. Fig. 5D: This in vivo experiment is crucial to the paper as it addresses the function of the PBP of Cyclin B3 in *Xenopus* oocytes. It demonstrates that the PBP of Cyclin B3 is essential to prevent the accumulation of XErp1 in meiosis I, thereby preventing a premature arrest in metaphase I. However, the western blots of endogenous XErp1 and Cyclin B3 are very faint. In Cyclin B3-depleted oocytes (CycB3Ab), XErp1 appears to be degraded 4 hours after GVBD, suggesting that exit from metaphase I is not abolished but only delayed. Moreover, the disappearance of ectopically expressed Cyclin B3 (WTΔ15 or PMΔ15) signals 4 hours after GVBD need explanation as it suggests that Cyclin B3 is degraded (which takes place during meiosis II). An additional marker beyond APC3 phosphorylation is required to confirm meiotic progression. A Cyclin B2 western blot would confirm that Cyclin B3-depleted oocytes, injected or not with CycB3PMΔ15, remain arrested in metaphase I as Cyclin B2 is not degraded after GVBD. A kinase assay would also be useful.

We thank the reviewer for this comment, which was also raised by reviewer #2. To rigorously address this point, we repeated the oocyte depletion/rescue experiment three more times. As shown in the **new** figure 2E, protein levels of XErp1 and cyclin B3 now show the expected behavior. Specifically, XErp1 accumulates prematurely (GVBD), and its levels continue to increase (GVBD+1h to GVBD+4h) in cyclin B3 depleted oocytes not expressing a rescue construct or expressing pocket mutant cyclin B3 (cycB3^{PMΔ15}). In

control depleted oocytes or cyclin B3 depleted oocytes expressing the WT rescue construct (cycB3^{WTΔ15}), XErp1 does not efficiently accumulate until GVBD+4h. Since these results were highly reproducible, we decided to exchange the old figure with the new one.

Seasonal variations in the speed oocytes progress through meiosis is a well-known phenomenon. Compared to the old oocyte batches, the new batches were more slowly in progressing through meiosis following progesterone treatment. This is for example evident in control depleted oocytes, which have not fully reached MII at GVBD+4h as indicated by the intermediate SDS-PAGE mobility of APC3 (more slowly than in prophase I (next lane to the right), but faster than in metaphase I). We realized that this circumstance makes it easier to determine the meiotic cell cycle stage of the oocytes under the different conditions. All oocytes unable to complete the first meiotic division, i.e., cyclin B3 depleted oocytes not expressing a rescue construct or expressing pocket mutant cyclin B3, display constantly upshifted APC3. In contrast, all oocytes that progress towards MII, i.e., control depleted oocytes or cyclin B3 depleted oocytes expressing WT cyclin B3, show intermediate SDS-PAGE mobility of APC3. As suggested by the reviewer, we included a cyclin B2 immunoblot, which further helps to determine whether oocytes are able to complete MI or not. Based on this novel data set, we thought that an additional kinase assay would no longer be useful to understand the conclusions drawn from this experiment.

It is puzzling that although oocytes were injected with both Flag-TRIM21 and Flag-Cyclin B3, the Flag western blot shows only one signal.

For our rescue conditions, we aimed to match the levels of the rescue construct cyclin B3^{Δ15} as closely as possible to those of the endogenous cyclin B3. Furthermore, we wanted to prevent that the rescue construct is continuously synthesized (even after completion of the first meiotic division), which would prevent oocytes to enter MII due to continuous XErp1 degradation. In our hands, the minimal amounts of mRNA present in IVT reactions are sufficient to support continuous protein translation. We, therefore, injected minimal amounts of IVT Flag-cyclin B3^{Δ15} into oocytes. In contrast, for TRIM-Away, we experienced that high levels of TRIM21 are required for efficient depletion and, therefore, injected rather large amounts of TRIM21 mRNA into oocytes. Hence, the Flag-TRIM21 signal is much stronger than the one of Flag- cyclin B3^{Δ15} in the WB shown in Fig. 5D.

7. Fig. EV1B and EV1C: Based on what is known about Cdk1-Cyclin B1-Cks1, the authors depleted Cks2 and found that Cyclin B3 triggers exit from metaphase II arrest independently of Cks2. It is unclear why the authors focused on Cks2 without analysing Cks1. In a paper from 2014 (10.1016/j.cub.2014.05.044), Cks1 was identified in metaphase II-arrested oocytes using proteomic, but at much lower concentrations compared to Cks2. This must be discussed by the authors.

We thank the reviewer for pointing this out. Our rationale to focus on Cks2 was based on the observation that mouse oocytes lacking CKS2 (CKS2^{-/-} oocytes) fail to progress past the first meiotic division⁹. To exclude that this phenotype is due to a role of Cks2 in targeting XErp1 for degradation, we analyzed if Cks2 depletion affects the Cdk1/cyclin B3-mediated degradation of XErp1. For the revised version of our manuscript, we describe the logic behind the experiment in more detail.

8. Material and methods:

- Include descriptions for antibodies against pThr (with the sequence of the phospho-site) and Separase.

We changed the material and methods section accordingly. Detailed information about the antibodies used in the manuscript can be found in the Reagents/Tools Table.

- Describe the phosphorylation protocol for XErp1NT and T97 peptides with the kinase that have been used.

We changed the material and methods section accordingly.

Minor comments

The manuscript should be edited for consistency and clarity. Unnecessary details, such as protein tags, should be omitted.

We adjusted the text accordingly removing unnecessary details like protein tags when appropriate.

Please, find below some examples for the Figures:

- Use consistent terminology (e.g., Cdc27/APC3).
- Correct labeling and include relevant details (e.g., "GVB" to "GVBD" in Fig. 5C, dashes missing between "Pro I" and "1h" in timelines of Fig. 5D).
- Maintain consistency in labeling across main and supplementary figures (e.g., Flag-Cyclin B3WT or Flag-Cyclin B3PM in all figures).

We thank the reviewer for pointing this out and adjusted the text accordingly.

References:

- 1 Elia, A. E., Cantley, L. C. & Yaffe, M. B. Proteomic screen finds pSer/pThr-binding domain localizing Plk1 to mitotic substrates. *Science* **299**, 1228-1231 (2003). <https://doi.org/10.1126/science.1079079>
- 2 Bouftas, N. *et al.* Cyclin B3 implements timely vertebrate oocyte arrest for fertilization. *Dev Cell* **57**, 2305-2320.e2306 (2022). <https://doi.org/10.1016/j.devcel.2022.09.005>
- 3 Hansen, D. V., Tung, J. J. & Jackson, P. K. CaMKII and polo-like kinase 1 sequentially phosphorylate the cytostatic factor Emi2/XErp1 to trigger its destruction and meiotic exit. *Proc Natl Acad Sci U S A* **103**, 608-613 (2006). <https://doi.org/10.1073/pnas.0509549102>

- 4 Liu, J. & Maller, J. L. Calcium elevation at fertilization coordinates phosphorylation of XErp1/Emi2 by Plx1 and CaMK II to release metaphase arrest by cytosolic factor. *Curr Biol* **15**, 1458-1468 (2005). <https://doi.org/10.1016/j.cub.2005.07.030>
- 5 Isoda, M. *et al.* Dynamic regulation of Emi2 by Emi2-bound Cdk1/Plk1/CK1 and PP2A-B56 in meiotic arrest of *Xenopus* eggs. *Dev Cell* **21**, 506-519 (2011). <https://doi.org/10.1016/j.devcel.2011.06.029>
- 6 Wu, Q. *et al.* A role for Cdc2- and PP2A-mediated regulation of Emi2 in the maintenance of CSF arrest. *Curr Biol* **17**, 213-224 (2007). <https://doi.org/10.1016/j.cub.2006.12.045>
- 7 Wühr, M. *et al.* Deep proteomics of the *Xenopus laevis* egg using an mRNA-derived reference database. *Curr Biol* **24**, 1467-1475 (2014). <https://doi.org/10.1016/j.cub.2014.05.044>
- 8 Hormanseder, E., Tischler, T., Heubes, S., Stemmann, O. & Mayer, T. U. Non-proteolytic ubiquitylation counteracts the APC/C-inhibitory function of XErp1. *EMBO Rep* **12**, 436-443 (2011). <https://doi.org/10.1038/embor.2011.32>
- 9 Spruck, C. H. *et al.* Requirement of Cks2 for the first metaphase/anaphase transition of mammalian meiosis. *Science* **300**, 647-650 (2003). <https://doi.org/10.1126/science.1084149>

Dear Thomas,

Thank you for submitting your revised manuscript. It has now been seen by all of the original referees. I apologize for the delay in getting back to you, it took longer than anticipated to receive the referee reports.

As you can see, the referees find that the study is significantly improved during revision and recommend publication. However, I need you to address the points below before I can accept the manuscript.

- Please address the remaining minor concern of referee #3.
- Please remove the Author Contributions section from the manuscript.
- Our production/data editors have asked you to clarify several points in the figure legends:
 - o Please note that the legend for figure 2e is mislabeled as 2d in the manuscript. This needs to be rectified.
 - o Please note that the figure title for figures EV 1-3 is not provided in the manuscript. This needs to be rectified.
 - o Please note that information related to n is missing in the legend of figure EV 2b.
 - o Although 'n' is provided, please describe the nature of entity for 'n' in the legend of figure 1f.
- Papers published in EMBO Reports include a 'synopsis' and 'bullet points' to further enhance discoverability. Both are displayed on the html version of the paper and are freely accessible to all readers. The synopsis includes a short standfirst summarizing the study in 1 or 2 sentences (max 35 words) that summarize the paper and are provided by the authors and streamlined by the handling editor. I would therefore ask you to include your synopsis blurb and 3-5 bullet points listing the key experimental findings.
- In addition, please provide an image for the synopsis. This image should provide a rapid overview of the question addressed in the study but still needs to be kept fairly modest since the image size cannot exceed 550 (width) x 300-600 (height) pixels.

Thank you again for giving us to consider your manuscript for EMBO Reports, I look forward to your minor revision.

Kind regards,

Deniz

--

Deniz Senyilmaz Tiebe, PhD
Senior Scientific Editor
EMBO Reports

Referee #1:

The authors have addressed my concerns during the previous round of review. Publication of this excellent study is recommended.

Referee #2:

I am satisfied with the revision/ The additional experiments have further improved the MS, which is now ready for publication. I suggest that this MS should be accepted by EmboReports.

Referee #3:

The authors have addressed all my concerns by performing new experiments and re-editing the paper. The manuscript has been greatly improved and is now much easier to read. I have one final minor comment: on page 4, the antibody against cyclin B3 used for Western blots is described as being raised against "aa 1-150", whereas in the Materials and Methods section, a fragment corresponding to "aa 1-110" is indicated. This should be corrected. I congratulate the authors for this new study, which can be published now.

All editorial and formatting issues were resolved by the authors.

Prof. Thomas Mayer
University of Konstanz
Biology
Universitätsstr. 10
Konstanz, BaWü 78457
Germany

Dear Thomas,

Thank you for submitting your revised manuscript. I have now looked at everything and all is fine. Therefore, I am very pleased to accept your manuscript for publication in EMBO Reports.

Congratulations on a nice work!

Kind regards,

Deniz

--

Deniz Senyilmaz Tiebe, PhD
Senior Scientific Editor
EMBO Reports

--
